# Multiple timescales of normalized value coding underlie adaptive choice behavior

Jan Zimmermann[1], Paul W. Glimcher[1,2] & Kenway Louie [1,2]

Adaptation is a fundamental process crucial for the efficient coding of sensory information. Recent evidence suggests that similar coding principles operate in decision-related brain areas, where neural value coding adapts to recent reward history. However, the circuit mechanism for value adaptation is unknown, and the link between changes in adaptive value coding and choice behavior is unclear. Here we show that choice behavior in nonhuman primates varies with the statistics of recent rewards. Consistent with efficient coding theory, decision-making shows increased choice sensitivity in lower variance reward environments. Both the average adaptation effect and across-session variability are explained by a novel multiple timescale dynamical model of value representation implementing divisive normalization. The model predicts empirical variance-driven changes in behavior despite having no explicit knowledge of environmental statistics, suggesting that distributional characteristics can be captured by dynamic model architectures. These findings highlight the importance of treating decision-making as a dynamic process and the role of normalization as a unifying computation for contextual phenomena in choice.

---

[1] Center for Neural Science, New York University, 4 Washington Place Room 809, New York, NY 10003, USA. [2] Institute for the Study of Decision Making, New York University, 4 Washington Place Room 809, New York, NY 10003, USA. Correspondence and requests for materials should be addressed to J.Z. (email: jan.zimmermann@nyu.edu)

Behaving organisms face complex and constantly changing environments, requiring the processing of vast amounts of information. However, information capacity is limited in neural systems, which face intrinsic physiological constraints such as finite numbers of neurons and bounded dynamic ranges in spiking activity. How such intrinsically limited neural systems represent near-limitless quantities of information has long been a fundamental question in neuroscience[1–3]. The efficient coding hypothesis proposes that neural systems exploit statistical regularities within their inputs, reducing redundancy to maximize the information carried in neural responses[1]. Information-maximizing coding strategies for capacity-limited systems have been extensively documented in sensory processing[2,4–8], but whether these principles extend to higher-order processes like decision-making is not fully known.

A key element in the efficient coding of sensory information is the contextual modulation of neural responses[7,9,10]. Under contextual modulation, responses to a stimulus depend on both the stimulus and the surrounding sensory context. Functionally, context effects can be grouped into two broad domains, driven by either spatial context or temporal context. In spatial contextual modulation, neurons encode stimulus-specific sensory information relative to surrounding contemporaneous sensory information[10,11]. In temporal contextual modulation, neurons adjust their sensitivity in response to recent sensory history to encode information relative to recent stimuli[12–15]. Such adaptation to stimulus history is a widespread feature of sensory physiology and perception. Critically, both spatial and temporal contextual modulation are thought to contribute to the efficient coding of sensory information, allowing the brain to minimize redundancy by accounting for spatial and temporal regularities in the sensory environment[2,7,10].

In contrast to sensory systems, less is known about how contextual modulation and efficient coding principles operate in the neural systems underlying economic decision-making. While context-dependent preferences have a prominent history in the behavioral choice literature[16–18], the responsible neural mechanisms have not yet been identified. Recent work suggests that spatial contextual modulation in value coding plays an important role in value representation and spatial choice effects. Neurons in monkey parietal and premotor regions represent the value of actions in a relative manner, normalizing firing rates to the spatial context defined by the value of the current choice set[19–21]. Similar relative value coding has been observed in human parietal cortex using electroencephalography[22] and functional magnetic resonance imaging[23]. This relative value coding is believed to be mediated by divisive normalization, a canonical neural computation prevalent in sensory brain areas. Normalization thus provides a unifying computational mechanism for contextual modulation in both sensory and decision processing. Importantly, normalized value coding also explains spatial contextual effects in value-driven decision-making[23,24], linking contextual modulation in neural coding to context-dependent choice.

While temporal context also controls value-coding activity, the underlying neural mechanism is less well understood and the link between mechanism and behavior remains unclear. Adaptation in value coding has, however, been observed in a number of brain regions involved in reward learning and value representation[25–28]. In the orbitofrontal cortex (OFC), an area implicated in the representation of option and outcome values[25,29–31], neurons represent value information independent of the current choice context[32] but dependent on the statistical structure of recent rewards. OFC value coding adapts to the range or variance of recent rewards[26,33], with less variable reward environments generating stronger value coding. Similar adaptive responses are exhibited by midbrain dopamine neurons[28], suggesting that adaptive value coding is a general feature of reward-processing in the brain. Such adaptation dynamically reallocates neuronal coding sensitivity to represent the most likely rewards, as determined by the recent reward history. A common theoretical assumption is that this sensitivity retuning should improve discriminability at the behavioral level; however, despite ample evidence for adaptation effects in sensory neurons, empirical evidence for adaptation-induced changes in perceptual discriminability is varied, subtle, and conflicting[13,34,35]. In decision-making, most previous experiments demonstrating adaptive value coding have not examined choice behavior[26,33] and the relationship between adaptation in neural coding and changes in empirical decision-making is unclear. While recent evidence suggests that choice behavior can vary with the mean and range of rewards[36–38] as well as the tendency to repeat choices[39] (hysteresis), little is known about the neural mechanisms responsible for adaptive changes in value coding and their potential role in choice behavior outside of an explicit learning context[40,41]. Thus a critical open question is whether and how value-based decision-making adapts to time-varying changes in the statistics of the reward environment.

Here we propose a general computational mechanism for adaptive value coding based on the principles of dynamic divisive normalization and examine the ability of this model to predict adapting choice behavior. Normalization is a canonical computational mechanism in which the activity of a neuron is divided by a summed common factor, usually comprising the summed activity of a neighboring pool of neurons[3]. Given the ubiquity of normalization in sensory modulation and its proposed role in economic spatial context effects, we hypothesize that normalization, as implemented in dynamic networks, also mediates temporal context effects in value coding and choice behavior. To test our hypothesis, we extend the normalization model to a fully dynamic form capable of capturing the effects of past reward information. This extended mechanism implements a dynamic, cascaded form of normalization[42] that can bridge spatial and temporal context effects in a generalized framework.

To test this proposed mechanism, we examine nonhuman primate choice behavior in environments with varying reward statistics. These findings show for the first time that economic choice behavior adapts to the local reward environment in a manner consistent with efficient coding theory. Furthermore, both the extent and across-session variability in adaptive choice behavior is captured by the dynamic normalization model, without any explicit knowledge of the underlying reward statistics. These findings indicate that normalization can account for both spatial as well as temporal context effects in choice and suggest that the underlying biophysical mechanism, even though unknown at a circuit level, may serve a common purpose: efficient coding of value representations.

## Results

**Dynamic normalization model of value adaptation.** To examine the neural basis of temporal context effects in choice, we implemented a dynamic divisive normalization model incorporating multiple timescales of integration. Divisive normalization is a neural computation widely observed in both early sensory coding and higher-order cognitive processes including visual attention, multi-sensory integration, and decision-making[3,24,43–45]. The characteristic feature of normalization is a divisive scaling, in which the input-driven response of a neuron is divided by the summed activity of a large pool of other neurons. This divisive scaling introduces an intrinsic relativity in neural coding and is thought to play a key role in contextual modulation[3,46]. If the divisive term includes information about immediate past inputs or neuronal responses, normalization has been demonstrated to characterize key features of adaptation in sensory processing[5,47].

Our dynamic model simulates the time-evolving activity of value-coding neurons using a set of normalization-based differential equations. In decision-making, dynamic normalization models can reproduce temporal aspects of normalized value coding seen in the monkey lateral intraparietal (LIP) area, including phasic responses at choice onset, time-varying value coding, and a delayed onset of contextual information representation[48]. This model features paired excitatory (rate coding, $R$) and inhibitory (gain control, $G$) neurons for each choice option, with divisive normalization implemented via recurrent inhibition: each gain control neuron sums network excitatory activity and inhibits its paired output neuron via divisive scaling. In this model, evolving neural activity is governed by a set of $N$ pairs of differential equations:

$$\tau \frac{dG_i}{dt} = -G_i + \sum_{j=1}^{N} \omega_{ij} R_j \tag{1}$$

$$\tau \frac{dR_i}{dt} = -R_i + \frac{V_i}{1+G_i} \tag{2}$$

where $i = 1,..,N$ are individual choice options, $V_i$ is the value of option $i$, $R_i$ and $G_i$ are the activity of the excitatory and inhibitory neurons representing option $i$, respectively, the parameters $\omega_{ij}$ weight the input $R_j$ to gain neuron $G_i$, and the timescale $\tau$ governs the timescale of system dynamics. Together, these equations capture the cross-option normalization that underlies spatial context effects on value coding as well as their underlying dynamics. Note that these dynamics are intra-trial dynamics (when $\tau$ is short as in our original model), describing activity

changes in a decision-related brain area over the course of a single choice. However, these fast dynamics are unable to capture processes such as adaptation that occurs over multiple trials, necessitating an extension to the previous model.

In order to capture longer timescale phenomena while retaining the ability of the model to capture fast firing rate dynamics, such as those in area LIP, we extended the dynamic normalization model with an additional circuit capturing inter-trial (slower) dynamics. Inter-trial dynamics are required because empirically observed adaptive value coding is sensitive to distributional parameters like the mean, range, and standard deviation of past rewards[25,26,33], which can only be estimated across multiple trials. Our model utilizes the same general circuit architecture as the previously published model[48] but incorporates cascaded circuits operating at long and short timescales, generating both slow and fast dynamics (Fig. 1a). As in the previous model, each fast network gain control neuron ($G_i^F$) computes a weighted sum of excitatory output neurons and inhibits its paired output neuron via divisive inhibition:

$$\tau^F \frac{dG_i^F}{dt} = -G_i^F + \sum_{j=1}^{N} \omega_{ij} R_j^F + \sum_{k=1}^{N} \alpha_{ik} R_k^S \tag{3}$$

$$\tau^F \frac{dR_i^F}{dt} = -R_i^F + \frac{V_i}{1+G_i^F} \tag{4}$$

However, in addition to fast circuit excitatory neurons ($R^F$), fast circuit inhibitory neurons also receive input from excitatory neurons ($R^S$) in an upstream slow circuit. Like the fast circuit, the

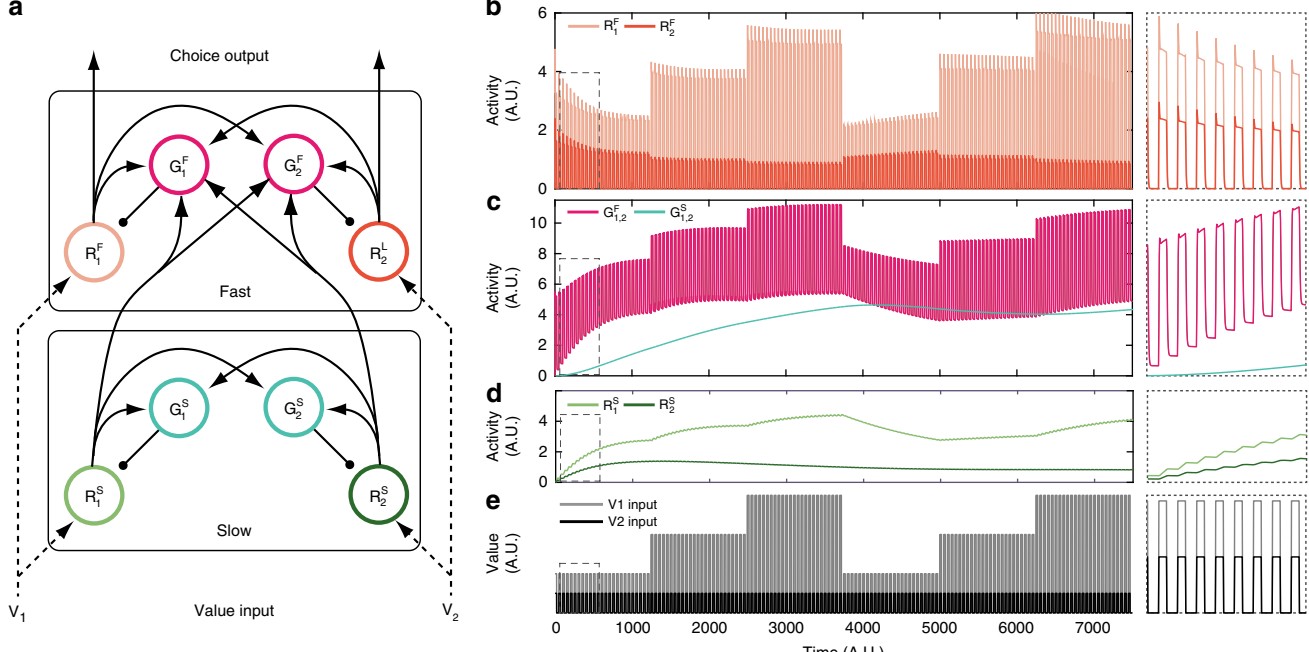

**Fig. 1** Cascaded normalization model and example model behavior. **a** Cascaded dynamic normalization model. Circuits are arranged into separate slow (bottom) and fast (top) subcircuits. Model units are either excitatory ($R$) or inhibitory ($G$); arrows depict excitatory and dots inhibitory connections. **b–e** The network behavior using a simple input structure emulating consecutive trials over time (**e**); panels on the right show an expanded view of a small time interval (dashed boxes). **b** Dynamic activity of the fast circuit excitatory units. These units show fast transients and subsequent sustained responses to value input in single trials. **c** Dynamic activity of fast (pink) and slow (teal) circuit inhibitory units. Note that, in the model architecture, inhibitory neurons within a subcircuit (fast or slow) act as a single pool and are plotted together. **d** Dynamic activity of slow circuit excitatory units. Over trials, slower adaptation-like effects within the slow circuit are propagated to the fast subcircuit. Contrasting the first and second half of the stimulation demonstrates the profound differences in value coding elicited by the temporal context despite having the exact same input structure (**b**). **e** Simulated value inputs. In this simulation, $V_2$ was kept constant while $V_1$ was changed across blocks; values were turned on during each trial and set to zero between trials

slow circuit comprises excitatory output neurons and inhibitory gain control neurons mediating a recurrent divisive inhibition but in this case at a much slower timescale (longer time constant):

$$\tau^S \frac{dG_i^S}{dt} = -G_i^S + \sum_{j=1}^{N} \beta_{ij} R_j^S \qquad (5)$$

$$\tau^S \frac{dR_i^S}{dt} = -R_i^S + \frac{V_i}{1 + G_i^S} \qquad (6)$$

Together, this system of differential equations (Eqs. 3–6) describes how the neural activity of each unit in the circuit changes over time as a function of activity levels and option values $V_i$. Both the fast and slow circuits utilize the same internal architecture: feedforward value inputs, lateral connectivity within a circuit, and recurrent inhibition. As in previous work[48], the fast circuit operates at a short timescale ($\tau^F$) to capture intra-trial dynamics, modeling value-coding activity in a brain area implementing choice. In contrast, the slow circuit operates at a long timescale ($\tau^S$) to capture inter-trial dynamics, modeling value responses that integrate responses over multiple trials and rewards. Critically, slow circuit activity modulates fast circuit activity via inputs to fast circuit gain control neurons, resulting in cascaded circuits operating at different timescales that influence both value coding and choice behavior via the output of the fast network.

**Example model behavior.** To demonstrate the multiple timescale characteristics of the cascaded model, we show model behavior in a simulation with values that change over long timescales (Fig. 1). In this simulation, we examined a two option network receiving information about two choice option values in individual trials. Within a block, the two option values were held constant; across blocks, the value of one option ($V_1$) stepped through 3 values (20, 40, 60 arbitrary units) twice, while the value of the other option ($V_2$) was constant (Fig. 1e). To signify individual trials, value inputs were turned on for a constant amount of time and switched off between trials. Time constants were set to 1 for the fast and 1000 for the slow subnetworks in arbitrary units of time.

Two important model characteristics are evident in the faster dynamic subcircuit of the network (Fig. 1b), which represents neurons in a value-coding output area (i.e., decision-related area). First, the fast dynamics in response to a single value input reflect previously studied dynamic normalization models and closely resemble aspects of value modulation in LIP area neurons[48]. Value-coding $R^F$ neurons in the fast network exhibit peak transients and subsequent steady-state plateaus during option presentation. Second, longer timescale changes in the system produce a change in the pattern of fast activity over time, despite constant patterns of input. This longer timescale adaptation is evident in changes in $R^F$ activity within each block. These adaptive changes in the fast network are driven by longer timescale changes in the slow network (Fig. 1c,d), not by changes in the inputs. Both excitatory ($R^S$) and inhibitory ($G^S$) units in the slow network operate at a longer timescale and integrate value information over multiple trials. Because inhibitory units in the fast network receive inputs from the slow network, their activity also exhibits longer timescale dynamics. This network architecture thus incorporates an interaction between the two dynamic components of the slow and the fast network.

These dynamics imply that the model can integrate information over multiple trials and thus reflect the statistics of the reward environment. In this model, we quantify choice formally by comparing the $R^F$ value-coding activity in the fast network for each of the options under consideration[48]. In the simulation shown in Fig. 1, $R^F$ activity representing the values of the two individual options changes over trials. Even with constant value inputs, the difference between these activities governing model choice also changes on a similar timescale (Fig. 1b, inset). Given these predicted neural and behavioral signatures of adaptation, we next examined whether the cascaded dynamic normalization model captures temporal context effects in empirical monkey choice behavior.

**Behavioral results.** To examine how temporal reward context affects empirical decision-making, we quantified choice behavior in monkeys performing a delayed two alternative forced choice task (Fig. 2a). In each trial, monkeys chose between options delivering different amounts and types of juice reward. Trials were organized into one of the two different kinds of blocks (narrow, wide; counterbalanced across sessions). Both blocks included identical test trials, which paired a fixed quantity of one juice type with varying quantities of a second juice type (Fig. 2b). The two juice types used were held constant for each animal. Within each block, these test trials always quantified choice behavior across a fixed set of value differences. In addition to these fixed test trials, each block included adapter trials, which paired varying quantities of the second juice type. Critically, the statistics of the adapter juice distribution differed between blocks of trials, exhibiting lower variance in the narrow block and higher variance in the wide block (while preserving a common mean). In each block, test trials and adapter trials were randomly interleaved; therefore, the statistical distribution of all presented rewards differed between blocks. This design allowed us to examine how choice behavior in test trials—which were identical between blocks—depended on the background reward statistics controlled by adapter trials.

We first examined how average choice behavior varied between blocks with different reward statistics. In sensory processing, the efficient coding hypothesis postulates a straightforward relationship between sensory statistics and neural stimulus coding[1,7]. Given a fixed dynamic range, neurons adapt to wider stimulus distributions with shallower firing rate response curves, a phenomenon that has been observed in value-coding areas of the brain[26]. We hypothesized that—if value-coding neurons adapt to narrow and wide reward distributions by changing the slope of their firing rate response curves—choice behavior for a fixed set of value differences should exhibit steeper choice curves in narrow reward contexts (Fig. 2c). For each session, we fit separate sigmoid functions to test trial choice data from the narrow and wide blocks (see Methods). In both monkeys, we observed a significant (Monkey H: $t(38) = 2.638$, $p = 0.012$; Monkey B: $t(29) = 2.199$, $p = 0.036$; $t$ test two tailed) change in average choice slope between the two adaptation conditions; this difference was also significant when tested in a non-parametric manner ($p < 0.05$; permutation testing). Choice slopes were on average higher in the narrow (smaller variance) condition compared to the wide (larger variance) condition (Fig. 3b). Notably, this change in choice stochasticity was not accompanied by a change in relative juice preference: neither animal exhibited a significant change between conditions in choice curve indifference points, the magnitudes at which juices are chosen with equal probability (Monkey H, $t(38) = 0.190$, $p = 0.850$; Monkey B: $t(29) = 0.603$, $p = 0.551$; $t$ test two tailed). We also found no significant difference in the root mean square error of the choice curve fits across block conditions (Monkey B: $t(29) = 1.405$, $p = 0.170$, Monkey H: $t(38) = 0.652$, $p = 0.518$; $t$ test two tailed); thus differences in model fit quality do not account for the adaptation-related difference in choice performance. This block-dependent

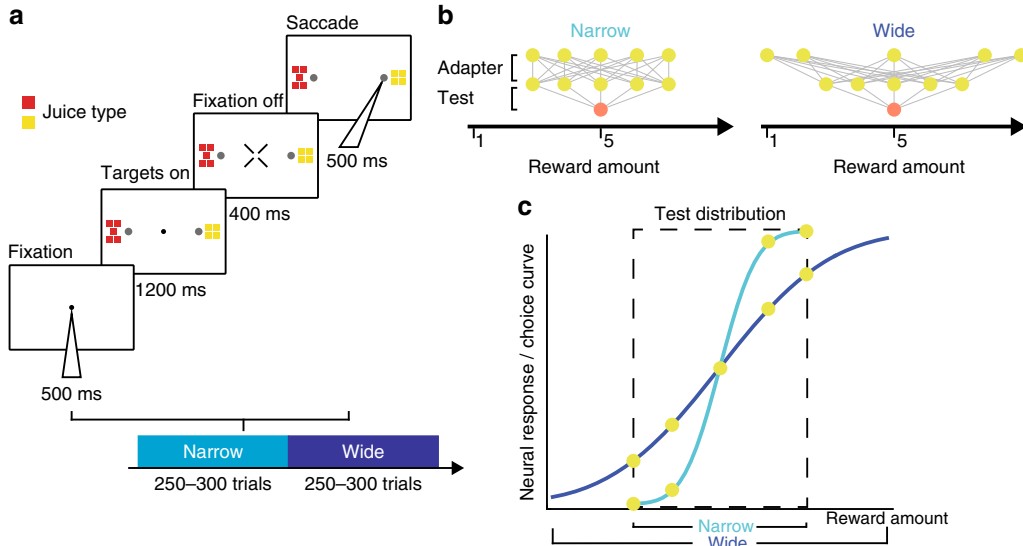

**Fig. 2** Choice task with varying background reward statistics. **a** Task design. Top, single trial timeline. Animals made saccadic choices between two rewards, with juice type and amount indicated by the color and number of squares, respectively. Bottom, block design. Individual test trials were presented in narrow or wide context blocks. **b** Trial types and block structure. Test trials were held constant between both conditions while adapter choice trials exhibited a larger standard deviation (same mean) of randomly presented reward options. **c** Illustrative data representing the hypothesized influence of a narrow and wide adapter distribution on neural coding and choice probability in the test distribution

difference in choice behavior demonstrates the influence of distributional context on choice stochasticity. Consistent with the predictions of efficient coding theory, adaptation to a larger distribution of value options leads to decreased choice performance on fixed test trial choices.

In addition to the significant average change in choice performance, adaptation effects demonstrated a large variability across individual sessions, each of which presented the adapters and test trials in a different random order. As evident in example daily sessions (Fig. 3a) and across all data (Fig. 3b), slope differences exhibited a range of adaptation effects: while most session differences occurred in the expected direction (narrow > wide), a minority exhibited either no difference (narrow ~ wide) or differences in the opposite direction (narrow < wide). This variability was unrelated to either varying indifference points or other potentially confounding behavioral measures. There was no significant relationship between narrow–wide slope differences and indifference point differences across sessions (Monkey H, $r(38) = 0.09$, $p = 0.56$; Monkey B, $r(29) = 0.20$, $p = 0.30$; Pearson correlation). Furthermore, a simple regression analysis showed no relationship between narrow–wide choice slope difference and various session variables (percentage of correct trials, experiment time, average saccade response time, mean received reward, standard deviation of received reward) across sessions (Monkey H, $F(38) = 1.68$, $p = 0.16$; Monkey B, $F(29) = 1.16$, $p = 0.35$; multiple linear regression). Thus the aggregate behavioral adaptation effect, which matches the prediction of efficient coding theory, masks a considerable variability in the effects of reward statistics on session-level data. Notably, because experimental reward distributions were identical across sessions, this session-by-session variability cannot be explained by the effect of average reward statistics. However, such variability might reflect the effect of more local reward statistics (i.e., the specific order in which adapters were presented), a possibility we next examined with the cascaded normalization model.

**Modelling adapting choice behavior**. To test the ability of the cascaded model to explain adapting choice behavior, we

examined whether it could reproduce two key features of the empirical data: (1) average choice performance across blocks, and (2) session-by-session variability in the extent of adaptation. Neither of these effects can be captured by static normalization models that ignore the trial history. To assess model performance, for each session the model was fed the identical trial sequence experienced by the monkey, trial-by-trial predicted choices were identified, and narrow and wide block choice curves were quantified. The inputs to the model thus comprised the exact experimental sequence of option values presented to a monkey, delivered with the same timing as in experimental sessions. Because the dynamic model utilizes differential equations, which evaluate the activity of a given unit as a function of both input and the activity of other units at each time step, both the magnitude and timing of value inputs control model activity and influence predicted choice behavior. In this approach, the temporal sequence of potential rewards encountered by the animals—but not the explicit underlying reward statistics—is available to the model; in order to adapt to block-wise reward statistics, the model must effectively extract this information from the dynamic sequence of trial values.

For each individual behavioral session, we determined predicted model choice on each trial and constructed probabilistic choice curves for the two block conditions (narrow, wide). As in our analysis of the monkey behavior, the difference in block-specific choice curves served as a measure of session-by-session adaptation in model-predicted choice behavior (Fig. 4a). Note that the dynamic model was not fit in a traditional sense to the choice data; the only free parameters in the model are the time constants $\tau^S$ and $\tau^F$. Only the ratio of these slow and fast time constants affects the behavior of the model. Rather than optimize parameter values, we examined model predictions at different fixed $\tau^S$ and $\tau^F$ ratios. Given that previous work suggests that fast normalization dynamics operate with a $\tau^F \sim 100$ ms[48], we assessed model predictions with $\tau$ ratios ($\tau^S/\tau^F$) ranging from 100 to 2500 consistent with a $\tau^S = 10–250$ s. At these slow timescales, the model incorporates information from several to many trials in the past (average trial length = 2.5 s).

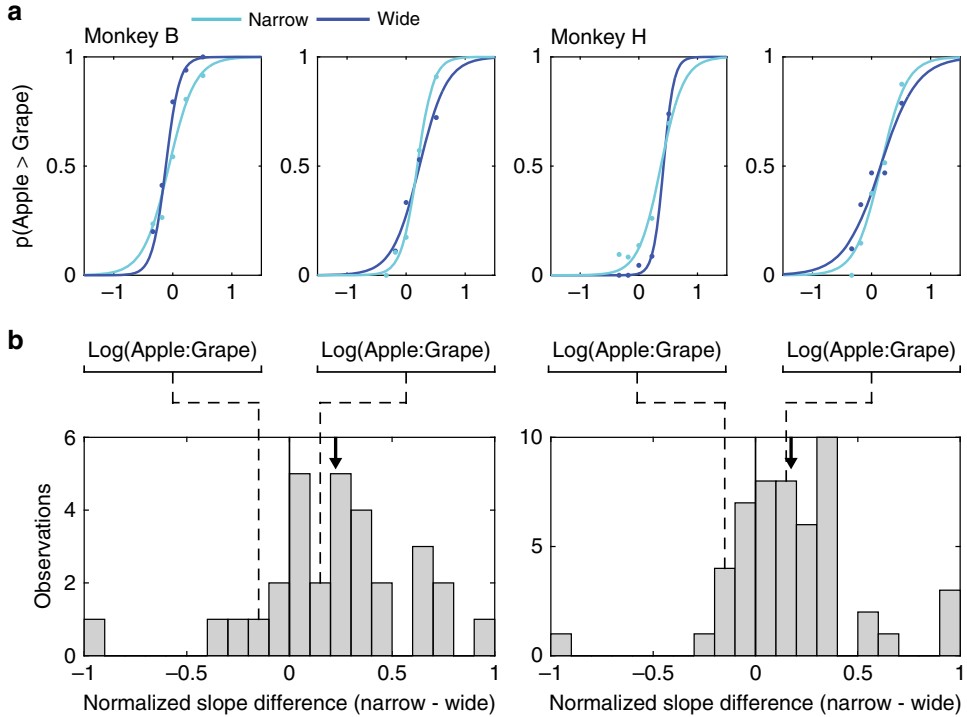

**Fig. 3** Individual choice data and aggregate behavioral performance. **a** Example single session monkey choice curves. Each choice curve is fit to data from single session test choice data from narrow (light blue line) and wide (dark blue line) blocks; dots show session average choice data. Data from two sessions in each animal are shown. Example sessions consistent with (right panel in each pair: narrow slope > wide slope) and contrary to the efficient coding prediction (left panel in each pair: narrow slope < wide slope) are displayed. **b** Normalized slope difference (narrow−wide) of all sessions performed by each animal. Left, Monkey B ($n = 30$ sessions); right, Monkey H ($n = 39$ sessions). A significant mean shift toward positive slope differences (narrow > wide) can be observed (arrow, mean slope difference)

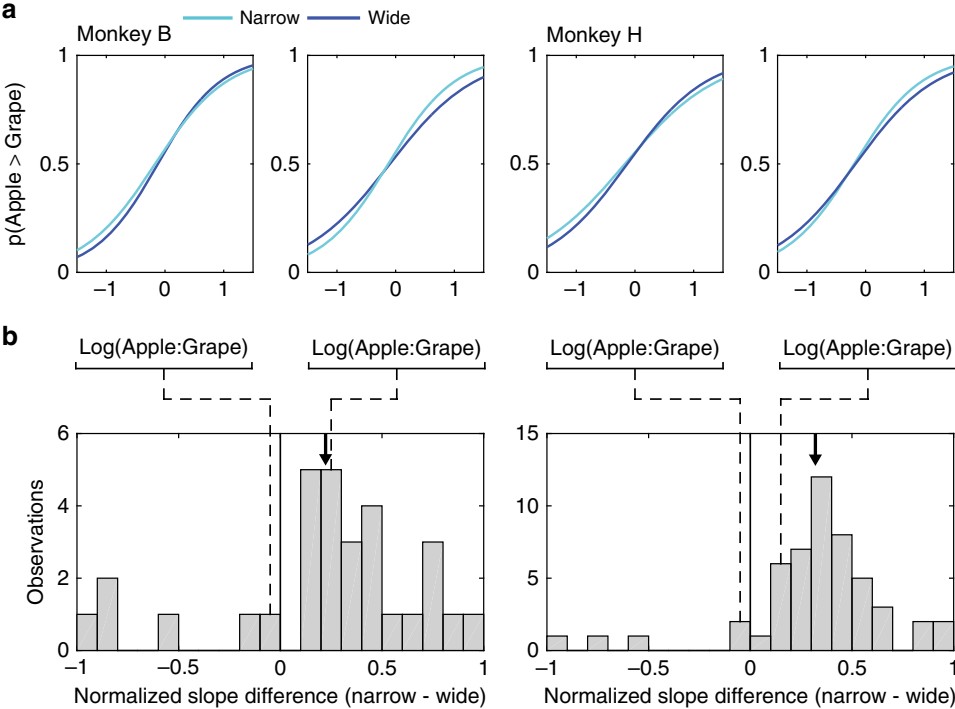

**Fig. 4** Individual model choice data and aggregate model performance. **a** Example single session model choice curves. Each choice curve is fit to model-predicted choices in single session test trials from narrow (light blue) and wide (dark blue) blocks. Model data from two sessions in each animal are shown. As in Fig. 3, example sessions consistent with (right panel in each pair: narrow slope > wide slope) and contrary to the efficient coding prediction (left panel in each pair: narrow slope < wide slope) are displayed. **b** Normalized slope difference of all sessions for the simulations. As in the empirical choice data (Fig. 3), a significant mean shift toward positive slope differences (narrow > wide) can be observed (arrow, mean slope difference). Example model-predicted choice curves in each monkey were based on optimal simulation timescales (Monkey B: $\tau^S = 75$ s; Monkey H: $\tau^S = 60$ s), as shown in Fig. 5b

We found that the cascaded dynamic normalization model captured both the mean adaptation effect and its session-by-session variability. Averaged across sessions (Fig. 4b), model predictions in both monkeys exhibit steeper choice curves in narrow versus wide block conditions ($p < 0.05$; permutation testing). The direction of model-predicted adaptation effects do not significantly differ from that observed in the behavioral data (monkey B: $\chi^2 = 1.36$, monkey H: $\chi^2 = 0.35$), indicating that the model and the monkeys exhibit analogous average responses to background reward statistics.

More importantly, as in the empirical data, model behavior exhibited across-session variability in adaptation. Figure 4a shows two example model-predicted choice curves in each monkey for the same sessions displayed in Fig. 3 (Monkey B: $\tau^S = 75$ s; Monkey H: $\tau^S = 60$ s). Across all sessions (Fig. 4b), model-predicted adaptation effects qualitatively matched observed adaptation effects, with slope differences consistent with the average effect (narrow > wide) and in the opposite direction (narrow < wide). The example choice curves and the distributions of slope differences indicate that model and animal behavior exhibit a similar pattern of across-session variability in adaptation. Moreover, the extent of adaptation in monkey and model behavior was significantly correlated across sessions (Fig. 5a; Monkey B: $r(29) = 0.453$, $p = 0.012$; Monkey H: $r(38) = 0.314$, $p = 0.046$; Pearson correlation). This correlation shows that the dynamic normalization model captures session-specific changes to the extent of adaptation: sessions that exhibited stronger adaptation in observed behavior generated stronger adaptation in model choice.

In order to differentially respond to distributional reward statistics, our normalization model relies on a slow circuit that integrates value information over multiple trials. To examine how model performance depends on the balance of slow ($\tau^S$) and fast ($\tau^F$) circuit timescales, we performed independent simulations of our model using a range of different $\tau$ ratio values (Fig. 5b). Model-predicted choice slope differences at each $\tau$ ratio were compared to empirical observations using correlational analyses, as described above. Significant correlations between model predictions and monkey behavior were only observed at intermediate $\tau$ ratios, corresponding to $\tau^S$ values of ~50–80 s in both animals (assuming $\tau^F = 100$ ms, based on previous work). These intermediate slow circuit integration times suggest that model dynamics must match the timescale of the relevant environmental statistics: at very fast integration times, the model is unable to capture across-trial reward statistics; at very slow integration times, the model is insensitive to across-block changes in reward statistics.

Finally, we examined which aspects of the reward environment contribute to model performance. In the primary analysis above, input to the dynamic normalization model comprised the exact sequence and timing of option values presented to the monkey in each session. To determine the contribution of reward sequence and timing to model performance, we examined model predictions with shuffled versions of the empirical data (see Methods and Fig. 6a). These shuffled data retained the choices associated with each presentation of options but varied either the sequence of presented options or their timing information. In the magnitude permutation, we shuffled the order of presented trials but retained the temporal characteristics of each session (trial and inter-trial interval (ITI) durations). In the ITI permutation, we shuffled the timing information but retained the sequence of presented trials. For each shuffled data set, model predictions and performance were determined at the best fitting $\tau^S$ for each animal, and the distribution of shuffled-data model performances quantified ($n = 1000$ repetitions each). Figure 6b shows the distribution of correlations for both magnitude and ITI

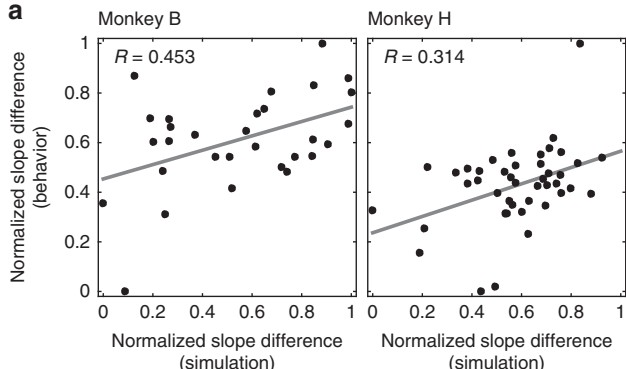

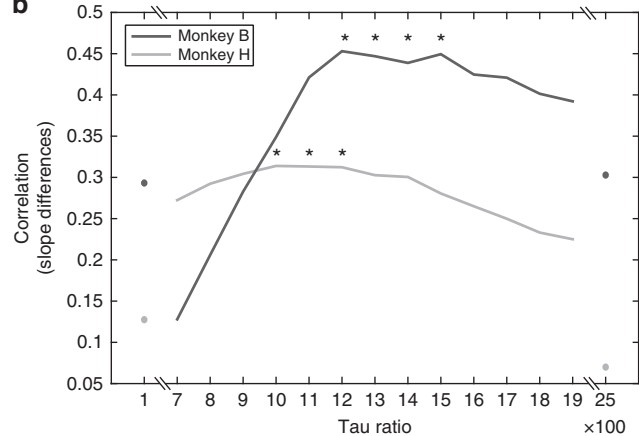

**Fig. 5** Across-session variability of adaptation strength captured by normalization model. **a** Correlation between strength of adaptation effect in monkey and model choices across sessions. Each point plots the difference in model choice stochasticity (narrow−wide) versus the difference in monkey choice stochasticity (narrow−wide) in a single session. Model performance evaluated using the best fitting tau ratio in each monkey. **b** Model performance at different slow circuit timescales. Each line shows correlation between model and monkey adaptation effects across all sessions (asterisk, Pearson's correlations significant at $p < 0.05$)

permutations. In both animals, our observed correlations (black line) are significantly different from the permuted null distribution ($p < 0.01$) for the magnitude shuffles and significantly different for the ITI shuffle in animal B ($p < 0.05$). The relatively small impact of timing information in model performance is expected given the experimental design: ITI variability was small (600–900 ms) compared to the timescale of the model slow circuit (~60 s). The ability of the ITI shuffles to decrease model performance is likely driven by post-error time-outs, which increased the interval between successive trials following aborted trials; consistent with this idea, the animal that exhibited a significant ITI shuffle effect also exhibited more time-outs on average (Supplementary Fig. 1). These findings show that the reward sequence, and to a lesser degree the precise timing, are necessary for the model to replicate empirical adaptation effects, suggesting that the influence of reward statistics on choice behavior may be implemented by the dynamical behavior of neural valuation circuits.

## Discussion

How intrinsically constrained neural systems efficiently represent the wide range of behaviorally relevant information is a fundamental question in neural coding. In sensory systems, spatial and

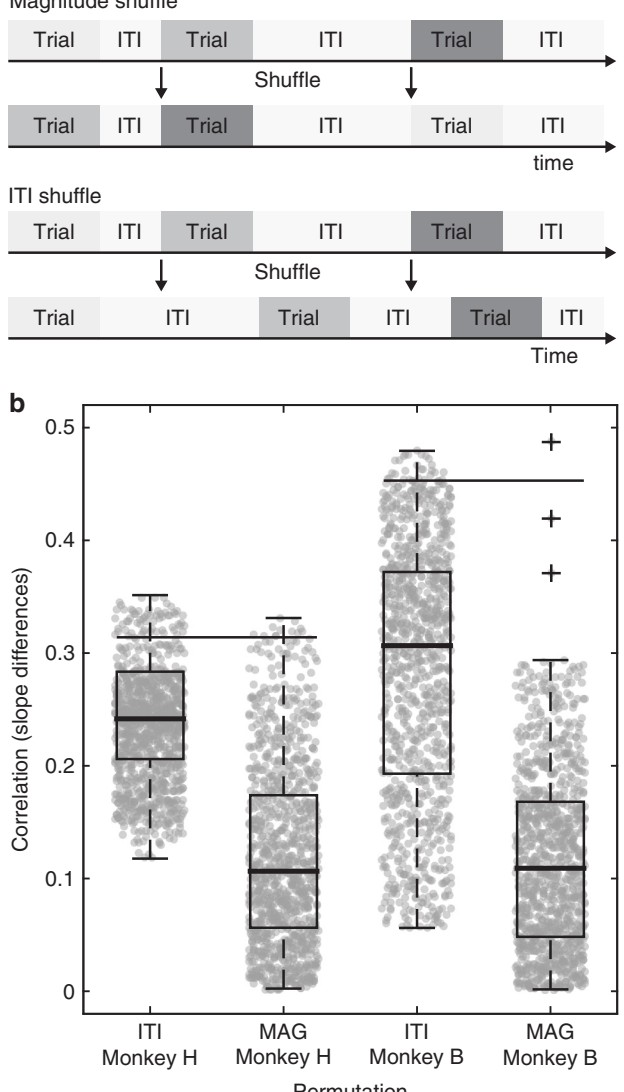

**Fig. 6** Permutation testing for monkey-model correspondence. **a** Illustration of permutation procedures. In the magnitude shuffle (top), the sequence of juice types and magnitudes option was permuted for each behavioral session. In the ITI shuffle (bottom), the sequence of reward magnitudes trial was kept intact but the exact timing of the presentation was shuffled by permuting the ITI intervals of each behavioral session (ITIs included timeouts that occurred following aborted trials). Each permuted dataset was then input to the dynamic normalization model. **b** Model–monkey correspondence on permuted data ($n = 1000$ permutations for each shuffle and monkey). Correlation determined in the same manner as in Fig. 5. Boxplot center line depicts the median, surrounding box 25th to 95th percentile, and shaded gray dots are underlying data distribution. Horizontal black lines show the correlation value obtained with the original non-permuted data

temporal contextual modulation of neural responses are thought to optimize neural coding for static and dynamic regularities in the sensory environment[2,3,10]. Importantly, these described coding changes predict behavioral changes at the perceptual level such as the tilt illusion and the motion aftereffect[10]. The extent to which these neural computational principles apply to value coding, and what influence they have on choice behavior, is

relatively less understood. Here we demonstrate a value adaptation effect on mean choice behavior in nonhuman primates. Consistent with efficient coding principles, the steepness of stochastic choice functions varies with the width of the recent value distribution. Notably, this adaptation effect displays a substantial amount of variability in both its magnitude and direction. We find that this adapting choice behavior can be explained by a dynamic value normalization model incorporating slow and fast timescale circuits. Dynamic normalization accounts for both the average adaptation effect as well as a considerable proportion of its variability, suggesting that value adaptation is driven by local reward changes rather than global reward statistics. While reinforcement learning processes in theory can produce sequential trial effects on valuation and generate choice stochasticity, the task used here is entirely deterministic in reward structure and a simple reinforcement learning model does not explain either the observed choice behavior or across-session variability in the extent of adaptation (Supplementary Fig. 2).

Adaptation is a characteristic feature of single neuron computation in sensory processing[7] and extends to value coding in reward-related brain areas, including midbrain dopaminergic nuclei and OFC[26,33,36]. Our results extend these neural findings to the behavioral domain, showing that—consistent with recent theoretical findings[36]—value-guided choice behavior also adapts to the recent reward environment. These adaptive changes in choice performance follow in principle from previously demonstrated value-coding changes, though establishing a definitive link between adaptation in value-coding activity and in choice behavior will require further study. In sensory processing, adaptation does not always reliably improve discriminability for stimuli similar to the adapter and often drives other changes in perceptual performance[34] (e.g., biases). While our results show a change in value-guided discriminability, adaptation to reward statistics could also produce other changes in choice behavior such as preference biases, particularly under different adaptation conditions (see Supplementary Discussion).

Regardless of the underlying biophysical mechanism, the slow temporal dynamics evident in our behavioral results emphasizes that the brain integrates information over real time rather than discrete trials. As our permutation tests demonstrate, merely shuffling the temporal structure reduces the predictive power of our model. This establishes that the actual temporal structure, and not merely the trial order, determines how reward history is integrated into an estimate of the reward environment. This finding is additionally supported by the fact that shuffling the temporal structure has a more detrimental effect on the behavior–model correspondence in the animal that has a larger variability in their temporal trial order.

Our modeling results also emphasize the importance of treating decision-making as a dynamic rather than a stationary process and demonstrate the power of a dynamic approach for explaining variability in choice behavior. The empirical adaptation effects we observed were variable across sessions despite identical average reward statistics, suggesting that adaptation responds to local, within-block fluctuations in reward environments. This variability in strength of adaptation was captured by a dynamic normalization model whose only input was the stream of observed rewards, indicating that behaviorally driven reward adaptation does not require explicit knowledge of distributional characteristics. A crucial element of this normalization model is the existence of two timescales of operation: a fast timescale in the choice behavior circuit and a slow timescale in the circuit integrating reward information over multiple trials. This dual timescale network is consistent with recent evidence that neurons exhibit a diversity of time constants[49,50]. In our data, optimal

slow circuit time constants were comparable between both animals (60–75 s); however, it is possible that environmental characteristics could influence the temporal integration process[51,52].

Our model follows from previous proposals of context dependent choice mechanisms[24,46,53] and can be seen as an approximation to a general mechanism for implementing flexible decision-making through context dependence. While previous work has demonstrated[19,48] that the fast dynamic circuit in our model corresponds well with electrophysiological characteristics of area LIP neurons, we remain open to the exact implementation and localization of the slow dynamic circuit. Our model implements a form of divisive normalization, a computation proposed to be a canonical cortical operation[3] that operates in many different neural processes, brain regions, and species. Our current model extends previously used normalization equations to incorporate both spatial and temporal context, further supporting the idea that normalization is a general neural computation. Many flavors of normalization models can potentially be conceived and the exact biophysical mechanisms are still under debate[3,54,55] but the importance of the implied computational mechanism lies in its simplicity and generality. Our approach extends this generality and reconciles adaptation-like processes under the umbrella of normalization (for limitations, see Supplementary Discussion).

In summary, we present here behavioral evidence demonstrating the effect of value adaptation on choice stochasticity. These adaptive changes in choice behavior are explained by a dynamic cascaded normalization model that captures session-by-session variability in the extent of adaptation and uncovers the behaviorally appropriate timescale integrating past reward information. Since decisions are rarely conducted in isolation, it is of crucial behavioral importance to understand how organisms adapt to the value context defined by the local temporal history. Establishing the dynamics of value coding and representation yields behavioral insights into choice behavior impossible to capture with traditional choice models.

## Methods

**Subjects**. Two male rhesus monkeys (*Macaca mulatta*) were used as subjects (Monkey H, 11.3 kg; Monkey B, 7.8 kg). All experimental procedures were performed in accordance with the United States Public Health Service's Guide for the Care and Use of Laboratory Animals and approved by the New York University Institutional Use and Care Committee. Experiments were conducted in a dimly lit, sound-attenuated room using standard techniques. Briefly, the monkeys were head restrained and seated in a custom-built Plexiglas primate chair that permitted arm and leg movements. Visual stimuli were generated using a liquid-crystal display (240 Hz) placed 67 cm in front of the animal. Monkeys' eye movements were monitored using the Oculomatic video-based system[56]. Monkey H's eye movements were also recorded using a scleral search coil, with horizontal and vertical eye position sampled at 600 Hz using a quadrature phase detector (Riverbend Electronics). Presentation of visual stimuli and juice reinforcement delivery were controlled with integrated software and hardware systems, controlled by a customized MonkeyLogic[57] package.

**Task**. Trained monkeys were offered a choice between two options differing in reward magnitude and juice type (delayed two alternative forced choice task). Monkeys were trained to fixate on a central fixation spot for 500 ms after which a choice display comprised of colored squares indicating the juice type and magnitude of the options was presented at 16° eccentricity. Monkeys had to maintain fixation for an additional 1200 ms until the fixation dot disappeared and a saccade toward one of the option targets could then be initiated. Monkeys' gaze had to reach the target within 500 ms and hold fixation on the target for an additional 400 ms to obtain the chosen juice reward (Fig. 2a). Time intervals between trials (ITI) were jittered between 600 and 900 ms. Juices were delivered through a multi-line juice tube. Each juice line was controlled by an independent solenoid valve. Routine calibrations were performed to match the juice quantity (~80 µl) to solenoid opening times. Magnitude of juice reward was realized by opening and closing (75 ms dead time) the solenoid according to the magnitude of the reward stimulus selected by the monkey.

Block of trials were presented that were composed of a mixture of adapter trials and test trials. Trial composition was assigned to be 60% adapter trials and 40% test trials, with individual trial identity determined randomly. In test trials, fixed in structure across all blocks, monkey chose between an unvarying reference reward (fixed reward magnitude and juice type) and one of the five variable rewards (Fig. 2b). These responses allowed us to plot the monkey's probability of choosing the reference reward as a function of the magnitude of the variable reward: a choice curve. What we systematically varied across blocks was the structure of the adapter trials. The structure was comprised of a narrow versus a wide standard deviation in the magnitude of randomly presented adapter trial options. We then examined the effects of the standard deviation of adapter variability on the slopes of these choice curves. Monkeys were required to complete both condition blocks within a single session. Thus, on a given testing day, animals were required to complete both one wide and one narrow block. Block order was randomized over recording days. Only days in which monkeys performed >230 trials per block correctly were included in the analysis. On average, animals performed 289 trials per block leading to an average of total of 578 trials per daily session. An additional accuracy criterion of 80% correct trials was used for data inclusion. A total of 9 days of data collection had to be discarded from analysis of which 5 days were due to poor animal performance (below 80%) and 4 days due to equipment failure. The average accuracy over both animals was 85%. Switches between blocks within a testing day were not overtly signaled to the animal. However, we note that, since offer quantities were unique between blocks, it is possible that the animal could identify block changes from this indirect signal.

**Data analysis and modeling**. Analysis of behavioral data and model performance used a total of 69 sessions (Monkey B, $n = 30$; Monkey H, $n = 39$). Only choice data from test trials (which were constant across conditions) were used for the following analysis. To examine the effect of adapter block identity on choice performance, we independently fit choice data from test trials in each block to a standard sigmoidal function ($y = 1/(1 + 10^{(x50-x) \times s)}$) and used the resulting slope of the choice curve as the parameter representing overall choice stochasticity. We then tested for systematic differences in choice stochasticity between the two behavioral conditions both on a mean aggregate level (all choices over days combined) or on a daily basis (all choices within daily conditions) and assessed the significance of the differences using permutation testing. We additionally ran a control regression analysis predicting condition choice stochasticity as a function of number of trials correct, condition completion time, and average response latency. For the modeling, we extended well-studied previous models of static divisive normalization to the temporal domain as a set of cascaded differential equations.

Our model uses a two-stage cascaded approach in which the incoming value information is first normalized with respect to the other concurrent value option via lateral inhibition. In a second stage, value information is normalized by a time discounted version of previously encountered value options. To estimate the predictive accuracy of this model, we used a two-fold approach. First, we individually solved the set of equations using the Runge–Kutta method for each behavioral time series. Behavioral time series were evaluated at a millisecond level with reward magnitudes and timing determined from the actual monkey experience in each session; note that model ITIs also included time-outs implemented following any aborted trials. Fixed parameters were used for evaluation ($\omega = 1$, $\beta = 1$, $\alpha = 1$ for all $i$ and $j$). These parameters were chosen to produce the simplest form of the model in which no a priori knowledge of excitation to inhibition weighting is known, baseline activity is zero, and inhibition is global to the cascade stage level. Excitatory and inhibitory $\tau$ values within a given circuit (slow, fast) were set to be equal. Previous work[48] has demonstrated that a network with equivalent excitatory and inhibitory time constants accurately characterizes value-coding activity in decision circuits; more broadly, standard mean-field approaches generally assume equivalent or approximately equivalent timescales within a network[58–60]. Temporal integration dynamics were modeled by explicitly setting a $\tau$-ratio between the fast and slow components of our network. We independently re-evaluated our model for a range of $\tau$-ratios spanning from short (5 s) timescales of about one trial to long (125 s) timescales encompassing many trials.

The resulting raw network activity was used to construct a probabilistic choice curve similar to the fitting of the behavioral results. The simple difference in modeled firing rate between the two presented options was used as a metric of choice output on a given trial. This means that, if the network consistently produces a large difference between concurrent presentations of the same options, the resulting choice probability becomes less stochastic. Variations in the firing rate differences over trials and time, as well as small differences in firing rates on the other hand, indicate larger stochasticity. In a final step, we aggregated the resulting modeled choice data and related the change in choice stochasticity over the experimental conditions to the changes observed in the actual behavioral data.

**Code availability**. The code used for data analysis and simulations is available from the corresponding author upon reasonable request.

**Data availability**. The data that support the findings of this study are available from the corresponding author upon reasonable request.

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

## Acknowledgements

We thank all members of the P.W.G. Laboratory, especially Kai Steverson for helpful comments and discussions. We also thank Rushell Dixon and Echo Wang for help with animal husbandry. This work was supported by grants from the National Institute of Health (R01MH104251 to K.L., R01DA038063 to P.W.G. and T32EY007136 to J.Z.).

## Author contributions

All the authors designed the experiment and wrote the manuscript. J.Z. collected and analyzed the data.

## Additional information

**Competing interests:** The authors declare no competing interests.

