## [Peer Review File · Nature Communications]

Reviewers' comments:

Reviewer #1 (Remarks to the Author):

Zimmerman J et al., Multiple timescales of normalized value coding underlie adaptive choice behavior

The authors present a theoretical model that can account for adaptation to the statistics of the reward environment. They use the model to account for changes in value-metric choice curves across conditions of a task where the reward distributions has low or high variance. They find that the slopes of the value-metric curves are steeper under low variance conditions, on average, and that this is accounted for by the model. The model also accounts for day-to-day variation in choice curves slopes, driven by fluctuations in the sequence of presented rewards. The time scale of the integration process that best accounts for behavior is approximately a minute.

This paper addresses an important issue in adaptive decision making, and makes predictions about changes in underlying neural tuning. The model is simple, clear and straightforward, and accounts well for changes in choice behavior. I have a few comments which may broaden the discussion.

Comments

1. Although the model development suggests that these results may be implemented within the cortex (with some mention of dopamine), thalamocortical interactions are anatomically well situated to implement this adaptation. Some features of the anatomical interaction between cortex and thalamus might also be appropriate. In addition, there could be a striatal component to the long time-scale adaptations. It would be worth discussing the possible contributions more broadly, instead of simply stating the cortico-centric view.
2. Although effects on the behavior here seem to be mediated by slow circuit dynamics (since only pairs of options are shown), there could be interactions between fast and slow circuit dynamics. The model as implemented postulates entirely separable effects.
3. There is another paper which has carried out an analysis of tuning in OFC looking at exactly this phenomenon, and I don't think it was referenced: Rustichini A et al., Optimal coding and neuronal adaptation in economic decision, Nat Comm, 2017.
4. The model is a low-pass filter. It's possible that animal's exposed to a large amount of training would learn that the statistics switch block-wise. The possibility that behavior might switch, instead of adapt slowly should at least be discussed.
5. I find it very surprising that permuting ITIs drops correlation so dramatically. Particularly given the very slow time constant of the filter, relative to the trial sequence. I would like to see the spectrograms of V_i , R_i_s and G_i_s (i.e. the slow R and G components like figure 1D), driven by the original and shuffled ITIs. It seems that, unless there is some structure in the data related to ITIs and the block type, this should have a minimal effect.

Reviewer #2 (Remarks to the Author):

This paper aims to describe choice variability by using a model that approximates a biophysically plausible mechanism, specifically adaption to local variability in rewards through divisive normalization over time. In general, I think this is a reasonable approach to undertake, but I have a number of questions and concerns about the study as it stands right now:

First, the manuscript omits a good deal of basic information that makes it difficult to evaluate. The

following information about the task and behavior should be added:

- How many blocks are included in the analyses?
- How many test trials were in each block?
- Did anything signal a block change?
- Were the types of juice used during adapter trials and test trials varied and counterbalanced, and if not why not? Were there only the 2 types of juice shown in the figure, or more?
- What was the overall accuracy on adapter trials for each monkey, and how many blocks were excluded from analysis by the imposed accuracy criterion?
- Did the behavior indicate that the monkeys took both juice type and amount into consideration in their choices? In other words, did they demonstrate a preference for one type of juice, and consistently prefer it in lower quantities than another, or is your sigmoid fitting comparisons of juice amount only?
- If they showed juice preferences, how consistent were these within and across blocks of the same type? for example, were there systematic changes in juice preference across trials that would suggest that they become satiated on one?

I also have a number of questions about the model estimation:

- What methods were used to optimize the tau parameters? That is, was it the correlation coefficient in 5B that was maximized? This would be rather unusual – why not something more standard like maximum likelihood?
- Was the optimization done on a block-by-block basis, or were blocks pooled, and did the authors consider whether the different conditions might have different optimal time constants? If it was done block-by-block, what does the distribution of tau parameters look like?
- Did the authors cross-validate their results, and how?
- For a paper focused on validating a model, it's notable that most model parameters were fixed at default values rather than optimized. I understand this was done for simplicity, but again this model is the entire focus of the paper. Is there precedent in the literature for doing this? Could the authors consider a global optimization of these parameters, or at least suggest how assuming these parameters might affect their results? Similarly, why were excitatory and inhibitory time constants assumed to be the same? Is there biological data to suggest this?

Conceptually, I think the conclusions would be strengthened if the authors contrasted their model with other methods that have been used to quantify choice behavior. For example, previous approaches to continuous variability have used temporal difference learning models (e.g. Bernacchia et al., 2011 or Walton et al., 2010, which should probably be cited in the manuscript). The key advance of this paper is proposing a better generative model for the behavioral observations, but it's hard to make that claim if other possibilities aren't formally assessed. In particular, it would be of interest to know what conditions or trial sequences (if any) are better captured by divisive normalization.

- Regarding the stats at the top of pg 12 – what data went into this analysis? are these from the slope distributions shown in Fig 3? Or was choice behavior pooled across blocks? Showing the average choice curves in a figure panel would be informative (also for the model predictions in fig 4).

- Fig 3A : Can the authors bring the individual dots in these panels in front of the fit lines, or otherwise make them more visible? And also clearly distinguish the dots belonging to each condition? This was particularly confusing in the first panel from monkey H, where only the points in the lower part of the plot jump out at first glance. Related to this, it appears that the curve for the 'narrow' block in this panel is fit to only points < 0 on the x-axis, so that a sigmoid function isn't a good fit at all. This raises the possibility that this is not an optimal description of behavior across all sessions, which might indicate that the monkey adopts different strategies in different blocks. For instance, this one looks like he primarily follows the rule "choose grape" rather than evaluating each offer. Trying to fit behavior that follows a heuristic like this with a model that assumes value-based choices could be misleading – does the model actually produce behavior like

this? These data are only from one block, so it's hard to know how prevalent this is in the results overall. The paper would be more sound if the authors quantify and report sigmoid fits, and consider only modeling those sessions in which sigmoid functions fit the data better than simpler alternatives. It would also be important to understand whether the fits vary systematically across conditions.

- Fig 5 shows normalized slopes (narrow – wide), but what about the un-normalized slopes? In other words, are both conditions fit equally well by the model?

Minor

- pg4, last paragraph – there is an unformatted reference
- Fig2 C does not have corresponding text in the figure legend. I assume this is illustrative/hypothetical data, not real data
- “test trials” are called “measurement trials” in the methods section. The authors should choose one terminology and use it consistently throughout
- The methods say “We additionally ran a control regression analysis predicting condition choice stochasticity as a function of number of trials correct, condition completion time, and average response latency.” Where are these results in the manuscript? I think they might be missing
- Padoa-Schioppa has looked in great detail at choice variability in very similar tasks. A few relevant papers that were not mentioned in this manuscript are Padoa-Schioppa, 2013 (Neuron) and Rustichini et al., 2017 (Nat Comm)

Reviewer #3 (Remarks to the Author):

The manuscript addresses temporal adaptation phenomena in value based decision making. The authors build on past work, including their own, showing that the neural representation of value adapts to recent reward history. They focus here on two aspects that are less well known: the circuit mechanisms of this reward adaptation, and the link between adaptive changes in the neural coding of value and changes in behavior. The authors do a great job connecting a vast literature on sensory coding, efficient coding, and value based decision making. The experimental results presented in the manuscript support the hypothesis that recent reward history affects choice in non-human primates. The authors also propose a dynamical circuit model that describes well the experimental results, suggesting that temporal statistics of rewards can be captured by dynamic circuit architectures that implement divisive normalization with two separate timescales, and offering a link between the empirical results and the literature on efficient coding. Overall the manuscript reads well, is original, and could influence thinking in the field, but I found some of the claims a bit confusing or based on inaccurate referencing to the literature.

One weakness is that the set up in the abstract and introduction emphasizes the lack of knowledge on the circuit mechanisms underlying of adaptation in the neural representation of value, but the manuscript is not really about circuits: even if the model is circuit-based, it is not meant to suggest a specific circuit implementation in the brain, it is offered simply as a description of the behavioral data. I think this is fine, but perhaps the focus on the abstract could be modified.

Another weakness is the proposal that adaptation to reward statistics controls choice sensitivity. The authors suggest this is in agreement with efficient coding ideas, but the literature on sensory adaptation and efficient coding is not uncontroversial: the effects of visual adaptation on discriminability are subtle, stimulus dependent, and varied (compare e.g. Gutnisky and Dragoi 2008; Zavitz et al. J Neurosci 2016; Adibi et al PLoS Comp Biol 2014; for reviews, Kohn and Solomon Curr Biol 2014; Snow et al F1000 2017), Also, contextual modulation (both in time and space) has been shown to induce perceptual biases much more often and more robustly than improvement in discriminability or sensitivity. This is also reflected in models linking contextual modulation, efficient coding, and divisive normalization (e.g. Schwartz et al JoV 2009). Related, on

page 4 "Such a retuning of neuronal sensitivities is thought to improve the discriminability (at the neuronal level) between options most likely to be encountered by an organism." References?

I was also concerned that the model captures only a small fraction of the variability in the data (Fig 5, correlation coefficients of .45 and .31). Based on these numbers, claims like "Dynamic normalization accounts for both the average adaptation effect as well as its variability," should be perhaps be tuned down a bit. More important, I would find it very helpful if the authors could comment on possible sources of the unexplained variability, and whether there are ways to incorporate it in the model.

I found the lack of alternative models problematic. For instance, are two separate populations/time constants something fundamental, or can this be achieved e.g. with a single recurrent network with heterogeneous time constants? If, as suggested in the Discussion, the author think there is a separation of time constants in different brain areas (LIP and OFC), how much flexibility would be needed for those areas to also capture other time scales different from those presented in these experiments? A smaller, related point is that the choice in the model of a single timescale for both R and G neurons should be motivated.

In model fits to data, how is the optimal ratio of timescales decided? Based on the peak of 5B? What value is actually used for figure 4? This should be stated more precisely.

Missing caption for Fig 2C

Typos

Fig 1 caption "Panels (B-E) demonstrates" remove *s*

Page 25, first line "(ode45)"?

Page 25, third-to-last line "han,d"

General response to the Editor and all Reviewers:

Before turning to the individual comments of the reviewers, we first want address a point raised by two of the reviewers and by the editor which we think is terribly important: Could we develop a direct method to compare our model's goodness of fit to other popular models? This is a great question, and in response to this suggestion, we spent a significant amount of time on this issue. We spent considerable effort attempting to develop a fitting method that would converge our model robustly and in finite time with our data and which would allow for an AIC/BIC comparison with existing models. The bad news is that we found this basically impossible and we wanted to take a half a page to explain why.

The problem, in a nutshell, is that our model is a dynamic model that uses a family of differential equations in multiple variables, unlike the discrete reinforcement models of other papers. What is critical here is that there is no analytic solution to our model because of its dynamic temporal structure – if you start to vary one of the parameters the model changes in complicated and unpredictable ways specifically because it is a set of coupled non-linear differential equations. That is why, as our readers may know, models like these tend to be addressed numerically rather than analytically.

Previous work from our lab (Louie et al. 2014) has demonstrated an analytic solution to the steady state behavior of static normalization models, and we have compared that steady-state model with other popular models in previous papers, but in this paper we have extended those models into the dynamic temporal domain as a family of nonlinear differential equations. This makes "fitting" the model quite a different problem than one encounters with discrete models that have just a few parameters. (The issue with this approach is the "ill defined" nature of multiple interconnected time constants all interacting together in a non-linear way.) In trying to develop a way to compare models we experimented with numerical approaches that made distributional assumptions on the relationship between time constants, but we have not identified a useful and practical approximation – basically we simply cannot get this model to "fit" in a convincing numerical fashion when the parameters are "free". It is conceivable and our hope that a general form of this network in a reduced form could be developed and used in the future, but that is a significant mathematical undertaking and a paper in its own right.

We also wanted to note, that the very different nature of the model explored here makes it even difficult to identify a competing model for direct comparison. The available models that act on different timescales are focused on learning and have no inherent properties to explain choice stochasticity as a function of the reward environment. They just are very different from the kind of model we use here (which is much more like neuron-level modeling in some ways).

The point of this (too long) soliloquy, is that while we have tried hard to re-conceptualize our approach in a way that made the paper about “model comparison” rather than a paper about “new approaches to choice dynamics,” we just couldn't find a way there. We believe that our current approach is highly unique in trying to explain choice stochasticity as a function of the reward environment mediated by temporal integration and the results have their own merit.

Reviewer #1:

Zimmerman J et al., Multiple timescales of normalized value coding underlie adaptive choice behavior

The authors present a theoretical model that can account for adaptation to the statistics of the reward environment. They use the model to account for changes in value-metric choice curves across conditions of a task where the reward distributions has low or high variance. They find that the slopes of the value-metric curves are steeper under low variance conditions, on average, and that this is accounted for by the model. The model also accounts for day-to-day variation in choice curves slopes, driven by fluctuations in the sequence of presented rewards. The time scale of the integration process that best accounts for behavior is approximately a minute.

This paper addresses an important issue in adaptive decision making, and makes predictions about changes in underlying neural tuning. The model is simple, clear and straightforward, and accounts well for changes in choice behavior. I have a few comments which may broaden the discussion.

Comments

1. Although the model development suggests that these results may be implemented within the cortex (with some mention of dopamine), thalamocortical interactions are anatomically well situated to implement this adaptation. Some features of the anatomical interaction between cortex and thalamus might also be appropriate. In addition, there could be a striatal component to the long time-scale adaptations. It would be worth discussing the possible contributions more broadly, instead of simply stating the cortico- centric view.

A1R1. We thank the Reviewer for this insightful comment, and agree that non-cortical structures may very well be involved in multiple timescale adaptation processes. Of course the Reviewer is entirely correct that subcortical-cortical interactions may play an important role in this phenomenon. We have expanded the discussion to specifically address this possibility. We have now

added a specific discussion of potential anatomical substrates of the normalization model subcircuits to the Discussion, touching on both cortical and subcortical brain regions:

“An unresolved question at this time is how slow temporal dynamics are biophysically implemented in neural circuits. One possibility is that these changes result from long term plasticity within decision circuits, akin to proposals for adaptation in the visual system¹⁰. Another possibility is that pooling the integrated output of value based activity and recurrent feedback may underlie this phenomenon. In our modeling efforts, we purposefully refrained from linking our model stages with particular brain areas; however our estimated slow temporal integration timescales match well with previous electrophysiological studies of adaptation in orbitofrontal cortex and our rapidly adapting stage relates well to data from parietal area LIP^{19,40}. However, it is possible that fast and slow timescale functions are served by a number of brain areas; alternatively, both timescales may be part of a single network capable of operating at a range of multiple timescales. Recent evidence of large-scale dynamical models based on connectivity data from tract-tracing experiments suggests a hierarchy of integrative timescales with sensory systems exhibiting brief transient responses and persistent long term activity in associative cortex^{44,45}. These findings suggest an unknown circuit mechanism that establishes long temporal receptive windows within prefrontal and temporal areas. One potential explanation for these differences can be regional differences in electrochemical composition of synapses⁴⁶. It is unclear if these synaptic changes are driven primarily within cortical regions or if potential thalamo-cortical projections regulate temporal integration⁴⁷, a potential mechanism underlying many theories of learning signals⁴⁸.”

2. Although effects on the behavior here seem to be mediated by slow circuit dynamics (since only pairs of options are shown), there could be interactions between fast and slow circuit dynamics. The model as implemented postulates entirely separable effects.

A2R1. We thank the Reviewer for this comment, which we fear suggests that we were not as clear about the limitations of our experiment and our model as we should have been. We hope the following clarification can address this limitation in the original manuscript: In the current experiment, slow rather than fast circuit dynamics necessarily drive adaptive changes, since reward variance only becomes evident when integrating over multiple presented trials. The Reviewer correctly points out that under some conditions within and between trial dynamics should interact, because our model is not entirely separable. Slow circuit activity alters behavior by controlling the fast circuit gain control. Thus, there is an interaction between the two dynamic components as the reviewer notes. If that were not the case, no adaptive effects on the current choice should be observable.

We agree that clarifying the interaction between the fast and slow circuits is crucial to conveying the function of the model, and have added the following explanatory text in the Results section:

“Because inhibitory units in the fast network receive inputs from the slow network, their activity – and their control of fast value coding units – also exhibit longer timescale dynamics. This network architecture thus incorporates an interaction between the two dynamic components of the slow and the fast network.”

3. There is another paper which has carried out an analysis of tuning in OFC looking at exactly this phenomenon, and I don't think it was referenced: Rustichini A et al., Optimal coding and neuronal adaptation in economic decision, Nat Comm, 2017.

A3R1. We thank the Reviewer for pointing out this oversight. During the initial submission of this manuscript, the paper from Padoa-Schioppa's group had not yet been published. We agree that this paper is relevant to our work is now been appropriately referenced in the Introduction and the Discussion.

Specifically, we address the Rustichini et al paper in the following revised text:

“While recent evidence suggests that choice behavior can vary with the range of rewards³⁶ as well as the tendency to repeat choices³⁷ (hysteresis), little is known about the neural mechanisms responsible for adaptive changes in value coding and their potential role in choice behavior.”

“Our results extend these neural findings to the behavioral domain, showing that – consistent with recent theoretical findings³⁶ - value-guided choice behavior also adapts to the recent reward environment.”

4. The model is a low-pass filter. It's possible that animal's exposed to a large amount of training would learn that the statistics switch block-wise. The possibility that behavior might switch, instead of adapt slowly should at least be discussed.

A4R1. The Reviewer makes a very interesting point about how animals may be able to detect changes in environmental statistics; we apologize for not discussing it explicitly in the previous manuscript. We acknowledge this is a possibility, and have added new text to the Methods:

“Switches between blocks within a testing day were not overtly signaled to the animal. However, we note that since offer quantities were unique between blocks, it is possible that the animal could identify block changes from this indirect signal.”

as well as new text to the Discussion:

“In our block design, neither our experimental animals nor our model required an overt cue or indication of the statistical change in the reward environment. However, it is possible that the monkeys learned to detect changes in environmental statistics and changed their decision behavior between contexts in a top-down manner. We note, however, that such a mechanism could not easily explain the across-session variability in the observed adaptation effect. Our data suggests a very high degree of sensitivity to the precise stochastic sequences of choices offered to the subjects, rather than to the block structure per se. Our shuffle analysis of the reward magnitudes within blocks further supports a continuous, rather than a change point-style process; it also indicates that the precise sequence of rewards and not the general identity of the blocks or statistics are the underlying driver for the adaptation effect we observed.”

5. I find it very surprising that permuting ITIs drops correlation so dramatically. Particularly given the very slow time constant of the filter, relative to the trial sequence. I would like to see the spectrograms of V_i , R_i s and G_i s (i.e. the slow R and G components like figure 1D), driven by the original and shuffled ITIs. It seems that, unless there is some structure in the data related to ITIs and the block type, this should have a minimal effect.

We thank the Reviewer for astutely noting that, given the slow time constant of the model, the ITI shuffle should have a reduced effect on model performance. This was a very helpful catch by the Reviewer and we are grateful for it. Given the issue raised in the last round of reviews, we went back and reexamined our previous results and discovered a coding error in the ITI shuffle analyses. We have reperformed our shuffle analyses and our revised results confirm the Reviewer’s intuition: while the reward magnitude sequence shuffle completely eliminates model performance (across-session correlation with extent of monkey adaptation), the ITI shuffle only partially decreases model performance (see revised Figure 6). This corrected ITI shuffle analysis is now presented in a revised Figure 6 as well as new text in the Results:

“Figure 6B shows the resulting distribution of shuffling for both order and intertrial temporal effects. In both animals, our observed correlations (black insert line) are significantly different from the permuted null distribution ($p < 0.01$) for the magnitude shuffles, and significantly different for the ITI shuffle in animal B ($p < 0.05$). The relatively small impact of timing information in model performance is not unexpected given the experimental design: ITI variability was small (600-900 ms) compared to the timescale of the model slow circuit (~ 60 s). The ability of the ITI shuffles to decrease model performance is likely driven by post-error time-outs, which increased the interval between successive trials following aborted trials; consistent with this idea, the animal that exhibited a significant ITI shuffle effect also exhibited more time-outs on average (Supplementary Figure 1). These findings show that the sequence of rewards, and to a lesser degree their precise timing, are necessary for the normalization model to replicate empirical adaptation effects, suggesting that the influence of reward statistics on choice behavior may be implemented by the dynamical behavior of neural valuation circuits.”

We further explore the Reviewer’s intuition in the Supplementary Material and Supplementary Figure 1, which address why the ITI shuffle partially disrupts model performance (and more so in one animal). We believe this occurs in our data because – in addition to the standard ITI intervals of ~ 750 ms, error trials resulted in timeouts of ~ 7300 ms. These longer intervals between successive trials are included in ITI lengths for both the non-shuffled and shuffled model evaluations. To further investigate this possibility, we plot histograms of the adjusted ITIs (time between two consecutive correct trials, which includes both assigned ITIs and post-error timeouts) as a supplementary figure (Supplementary Figure 1). These figures demonstrate two main points: First there is trivially a periodicity to the accumulated adjusted ITI for both animals, meaning that animals sometimes miss multiple trials consecutively leading to longer time outs. Second, the animal that exhibited a larger detriment for the ITI shuffle analysis has overall longer accumulated adjusted ITIs. This finding provides a possible explanation for why the ITI shuffle analysis affects our model-behavioral correlations, including why the shuffle effect was stronger in one animal in comparison with the other. Again, much thanks to the Reviewer for bringing up these issues!

Reviewer #2:

This paper aims to describe choice variability by using a model that approximates a biophysically plausible mechanism, specifically adaption to local variability in rewards through divisive normalization over time. In general, I think this is a reasonable approach to undertake, but I have a number of questions and concerns about the study as it stands right now:

First, the manuscript omits a good deal of basic information that makes it difficult to evaluate. The following information about the task and behavior should be added:

General response R2. We would like to thank the Reviewer for pointing this out and we apologize for any lack of clarity. We have now thoroughly revised our manuscript to add the important details of basic information suggested. Please see the text below for specific changes.

- How many blocks are included in the analyses?

A1R2 Two blocks were presented to the animal on each day leading to one transition; the order of block presentation (narrow->wide, wide->narrow) was randomized across daily sessions. We have expanded the current language "Monkeys were required to complete both condition blocks within a single session." to be more clear. In the Methods section (under *Task*), the revised manuscript now reads:

"Monkeys were required to complete both condition blocks within a single session. Thus, on a given testing day, animals performed both one wide and one narrow block."

Overall, we examined empirical choice behavior and model performance in 69 sessions (pairs of blocks), 30 in Monkey B and 39 in Monkey H. This is now stated in the revised Methods section (under *Data analysis and modeling*):

"Analysis of behavioral data and model performance used a total of 69 sessions (Monkey B, $n = 30$; Monkey H, $n = 39$)."

- How many test trials were in each block?

A2R2 The previous manuscript language stating “Only days in which monkeys performed more than 230 trials per block correctly were included in the analysis.” has been expanded to state the actual mean trial numbers performed more clearly. In the Methods section (under *Task*), the revised manuscript now reads:

“Only days in which monkeys performed more than 230 trials per block correctly were included in the analysis. On average animals performed 289 trials per block leading to an average of total of 578 trials per daily session.”

- Did anything signal a block change?

A3R2 We apologize for not having been more clear on this as it was also pointed out by Reviewer 1. There was no explicit signal for a block change, and the number of trials per block varied slightly across days. We have revised the Methods section (under *Task*) to explicitly state this:

“Switches between blocks within a testing day were not overtly signaled to the animal.”

However, as pointed out by Reviewer 1, considering the overtrained nature of these animals it is conceivable that an animal could use adaptor trials (some of which have unique quantities when compared between blocks) as a signal for block identity switches. We have also added the following clarification to the Methods section:

“However, we note that since some offer quantities were unique between blocks, it is possible that the animal could identify block changes from this indirect signal.”

- Were the types of juice used during adapter trials and test trials varied and counterbalanced, and if not why not? Were there only the 2 types of juice shown in the figure, or more?

A4R2. The Reviewer is entirely correct in stating that only two juice types were ever used for a given animal. (The two juice types were different for the two animals since we calibrated the juices with

dilution to yield approximate isopreference). One type of juice was used for the adaptor trials (which presented two quantities of that type of juice) and two types of juice (the one from the adaptor trials, plus an additional juice type) were used for the test trials (which presented various paired quantities of the two juice types).

Importantly, the juice types and their assignments (adaptor trial type, test trial type) were not varied across blocks or sessions because we specifically wanted to examine the effect of reward distributions while holding juice type constant. Under our experimental design, for a given animal, test trials always paired the same types and amounts of juices in different blocks and in different sessions. Thus, any difference in behavior across blocks (adaptation effect) or across sessions (variability in adaptation effect) would be independent from changes in juice type. Adding different juices on a daily basis would have possibly confounded our analyses and/or potentially added noise to the behavior.

We thank the Reviewer for the point of clarity, and have now added a clarification to the Results section:

“The two juice types used were held constant for each animal tested”.

- What was the overall accuracy on adapter trials for each monkey, and how many blocks were excluded from analysis by the imposed accuracy criterion?

A5R2 We thank the Reviewer for raising this point, and apologize for our lack of clarity. As stated above, only days in which monkeys performed more than 230 trials per block correctly were included in the analysis. In the previous version of the manuscript, we mistakenly omitted the additional accuracy criterion, which excluded a small number of sessions. Accuracy was defined as animals performing at least 80% of the trials correctly. Errors were defined as any mistake made by the monkey during task execution, including not acquiring initial fixation, fixation breaks, early saccade initiation, and saccade endpoint errors; note that in order to not bias subsequent choice-based analyses, errors were not classified based on monkey choice performance. Out of all testing days, 2 sessions from animal B and 7 sessions of from animal H were excluded for poor performance (<80%). We note that in animal H, 3 of those days were related to an eye coil detector malfunction and 1 of those days to a solenoid failure; thus, poor performance independent of equipment failure was therefore observed on 5 days total for both animals.

We have expanded the Methods section to include the criterion and mention the excluded data:

“On average animals performed 289 trials per block leading to an average of total of 578 trials per daily session. An additional accuracy criterion of 80% correct trials was used for data inclusion. A total of 9 days of data collection had to be discarded from analysis of which 5 days were due to poor animal performance (below 80%) and 4 days due to equipment failure. The average accuracy over both animals was 85%.

- Did the behavior indicate that the monkeys took both juice type and amount into consideration in their choices? In other words, did they demonstrate a preference for one type of juice, and consistently prefer it in lower quantities than another, or is your sigmoid fitting comparisons of juice amount only?

A6R2. Juice dilution was made on a per-individual basis in order to keep the two rewards roughly isopreferent. As evident in the sigmoid fits, however, the monkeys did take both juice type and amount into consideration in some sessions, consistent with an economic decision of the type studied previously by Padoa-Schioppa and colleagues. Specifically, the sigmoidal fitting process takes into consideration both amount and type by fitting both a slope and a point of indifference; the fact that the indifference point of the curves (the point at which the integral over and under the curve are equivalent) is slightly shifted away from equal magnitude options ($\log(\text{juice A} : \text{juice B}) = 0$) in some sessions demonstrates that monkeys did take both types of information into consideration. However, there was: (1) no difference in indifference points between block conditions (narrow vs. wide), and (2) no relationship between indifference point differences between blocks in a given session and slope differences. We have now added text to state these points (see answer A7R2 below).

- If they showed juice preferences, how consistent were these within and across blocks of the same type? for example, were there systematic changes in juice preference across trials that would suggest that they become satiated on one?

A7R2. When looking at the indifference points of the choice curves between blocks, no significant difference could be observed for either animal (Animal B, $t = 0.6086$, $p = 0.5475$, Animal H, $t = 0.1842$, $p = 0.8549$), demonstrating that juice preference does not systematically change between testing block conditions (narrow vs. wide). Next we checked whether small variations in the indifference point co-vary with changes in slope across testing days. This test also came out non

significant (Animal H, $r = 0.0964$, $p = 0.5592$, Animal B, $r = 0.1988$, $p = 0.2924$), confirming that the slope differences we observed and reported in the manuscript are indeed only related to changes in choice stochasticity.

We thank the Reviewer for raising these points, and now include specific text and results regarding preference and indifference points in the revised Results:

“Notably, this change in choice stochasticity was not accompanied by a change in relative juice preference: neither animal exhibited a significant change between conditions in choice curve indifference points, the magnitudes at which animals chose a juice type with equal probability (Monkey H, $t(38) = 0.18$, $p = 0.85$; Monkey B: $t(29) = 0.61$, $p = 0.55$).”

and:

“This variability in choice stochasticity difference was unrelated to either varying indifference points or other potentially confounding behavioral measures. There was no significant relationship between narrow-wide slope differences and indifference point differences across sessions (Monkey H, $r(38) = 0.09$, $p = 0.56$; Monkey B, $r(29) = 0.20$, $p = 0.30$). Furthermore, a simple regression analysis showed no relationship between narrow-wide choice slope difference and various session variables (percent correct trials, experiment time, average saccade response time, mean received reward, standard deviation of received reward) across sessions (Monkey H, $F(38) = 1.68$, $p = 0.16$; Monkey B, $F(29) = 1.16$, $p = 0.35$).”

- What methods were used to optimize the tau parameters? That is, was it the correlation coefficient in 5B that was maximized? This would be rather unusual – why not something more standard like maximum likelihood?

A8R2. We apologize for any confusion in our presentation. First, we note that model behavior is governed by the ratio of τ values (τ^S/τ^F) rather than their individual values, since the ratio of timescales governs the relative influence of slow circuit activity on fast circuit activity. However, as discussed in the manuscript, the original fast circuit model was developed to model LIP firing rates and previously shown to match empirical neural activity at a $\tau^F \sim 100$ ms (Louie et al., *J Neurosci*, 2014), allowing us to interpret τ ratio values in equivalent τ^S values.

To address the Reviewer's point, the τ parameters were not explicitly maximized. In our approach, we chose for two reasons to demonstrate that the model can predict behavioral data (both average blockwise adaptation effects and variability across sessions) at reasonable τ values rather than fit optimal τ values. First, our goal was to generally demonstrate that this type of model can capture aspects of behavioral adaptation, rather than optimize the ability of the model to do so. Second, the model relies on a differential equation analysis across long streams of time-locked behavioral events. This computation is extremely compute intensive even for a single τ ratio, making it difficult to fit parameters. Thus, we opted to simulate model performance across a range of tau values to qualitatively examine how the model-monkey correspondence varies with τ values (Fig. 5). The parameters were chosen based on timescales suggested by the literature (Kobayashi et al. 2011, Padoa-Schioppa 2013). Note that this analysis was primarily performed not to identify an optimal parameter value but to: (1) examine if our general estimates of the timescale of adaptation aligns with that previously reported in the literature, and (2) reassure the reader that model performance is not artifactually high at all parameter values.

- Was the optimization done on a block-by-block basis, or were blocks pooled, and did the authors consider whether the different conditions might have different optimal time constants? If it was done block-by-block, what does the distribution of tau parameters look like?

A9R2. We point out that there was no optimization performed; instead the analysis involved: (1) setting a τ ratio, (2) computing model predictions at that value, and (3) quantifying the correlation between model and monkey choice slopes across sessions. Since our interest lies in explaining the variability of the slope differences across testing days, simulations for a given τ ratio were done across the entire dataset. We have revised the Results text to read:

“Note that the dynamic model was not fit in a traditional sense to the choice data; the only free parameters in the model are the time constants τ^S and τ^F . Only the ratio of these slow and fast time constants affects the behavior of the model. Rather than optimize parameter values, we examined model predictions at different fixed τ^S and τ^F ratios.”

- Did the authors cross-validate their results, and how?

A10R2. Because the τ ratio parameter was not optimized (in fact, all parameters were fixed, as noted above), no cross-validation was performed. The parameter is however extremely similar between both animals.

- For a paper focused on validating a model, it's notable that most model parameters were fixed at default values rather than optimized. I understand this was done for simplicity, but again this model is the entire focus of the paper. Is there precedent in the literature for doing this? Could the authors consider a global optimization of these parameters, or at least suggest how assuming these parameters might affect their results? Similarly, why were excitatory and inhibitory time constants assumed to be the same? Is there biological data to suggest this?

A11R2. The Reviewer is certainly correct that we focus on examining model performance with fixed parameters rather than optimizing the parameters themselves. We want to stress that in this paper we are not trying to make a point of which model (or set of parameters) is necessarily the best model, but rather if a model of the type we describe can capture aspects of across session variability at all. Many of the parameters used would need to be optimized if the data we captured were on a finer scale, for example if we were comparing model unit activations with actually recorded neural data. This is indeed a target of our future work, and neurophysiological recordings in animals performing the experimental task described here are underway. These upcoming papers will employ exactly the strategy proposed by the Reviewer.

The Reviewer also makes an interesting point about the excitatory inhibitory time constant ratio that we kept equal. The immediate motivation for assuming a single timescale is that we used equivalent excitatory and inhibitory time constants in a previous, single-stage version of the model (Louie et al, 2014). In that paper, a single fast-timescale circuit was shown to accurately describe multiple aspects of LIP value coding activity, and in our extension to a two-stage circuit here we kept the same convention. However, more generally, our rate model is similar in construction to a number of standard models taking mean-field approaches (Wilson-Cowan model and variants; see Wilson & Cowan, 1972, Humanski & Wilson 1992; also see Cowan et al 2016 for review). These models generally assume that excitatory and inhibitory time constants are either equivalent or closely related. To clarify this motivation, we have revised text in the Methods section as follows:

“Previous work⁴⁵ has demonstrated that a network with equivalent excitatory and inhibitory time constants accurately characterizes value coding activity in decision circuits; more broadly, standard mean-field approaches generally assume equivalent or approximately equivalent timescales within a network⁶¹⁻⁶³.”

For example, previous approaches to continuous variability have used temporal difference learning models (e.g. Bernacchia et al., 2011 or Walton et al., 2010, which should probably be cited

in the manuscript). The key advance of this paper is proposing a better generative model for the behavioral observations, but it's hard to make that claim if other possibilities aren't formally assessed. In particular, it would be of interest to know what conditions or trial sequences (if any) are better captured by divisive normalization.

A12R2. We thank the Reviewer for bringing up this point about other approaches to variability in behavior, and find it important to discuss the differences between our and previous approaches. To make sure we are discussing the correct papers referenced by the Reviewer, Bernacchia et al. 2011 refers to [Bernacchia, A., Seo, H., Lee, D., & Wang, X. J. (2011). A reservoir of time constants for memory traces in cortical neurons. *Nature neuroscience*, 14(3), 366-372.] and Walton et al. 2010 refers to [Walton, M. E., Behrens, T. E., Buckley, M. J., Rudebeck, P. H., & Rushworth, M. F. (2010). Separable learning systems in the macaque brain and the role of orbitofrontal cortex in contingent learning. *Neuron*, 65(6), 927-939.]. We agree with the Reviewer that these are relevant citations; we now cite the Bernacchia paper and Walton papers in the revised manuscript.

“These findings suggest an unknown circuit mechanism that establishes long temporal receptive windows within prefrontal and temporal areas⁴⁸.”

“While recent evidence suggests that choice behavior can vary with the range of rewards³⁶ as well as the tendency to repeat choices³⁷ (hysteresis), little is known about the neural mechanisms responsible for adaptive changes in value coding and their potential role in choice behavior outside of an explicit learning context³⁸”

Bernacchia et al. examined monkeys performing a competitive game (matching pennies), in which their behavior was influenced by past rewards and actions and could be fit with standard reinforcement learning (RL) models. Their important neural finding was that neurons (in ACCd, DLPFC, and LIP) showed a memory of past rewards, with increases/decreases in activity following reward/no-reward events; strikingly, these reward memory traces ranged widely across the neural population, from hundreds of milliseconds to tens of seconds. They find that a reward-driven recurrent network model can reproduce the power-law distribution of empirically observed time constants, but this model did not produce choice behavior and was not compared to monkey decision making or RL-model fits of model behavior. Walton et al. examined the behavior of OFC lesioned monkeys performing different variants of a three-armed bandit task. Their primary finding was that OFC lesions disrupt reversal learning, but primarily through disrupting the ability to assign outcomes to the correct previous choices, this driving an increased stochasticity in behavior following OFC disruption. The Walton et al. paper employed a RL model to determine subjective values for analysis, but not to characterize changes in behavior or levels of stochasticity.

We agree with the Reviewer that testing our dynamic normalization model against other models is ultimately important, but the Bernacchia and Walton approaches provide difficult points of comparison for several reasons. First, there are significant differences between our experimental task and the tasks in the Bernacchia and Walton papers. Both of the previous tasks involve explicit learning scenarios where animals use past information (rewards and/or choices) to update their behavior; this introduces a degree of nonstationary behavior that is different than in our experiment, where the reward contingencies in each trial are explicitly and unambiguously signaled by the cues. Unlike the Bernacchia and Walton tasks, in our experiment, there is no reason for the monkeys to change their behavior from trial to trial or between block conditions. In addition, both of the previous experiments employ probabilistic outcomes that drive additional choice stochasticity; in particular the lack of rewards on some trials is an important determinant of neural responses and behavior in the Bernacchia results. Second, neither the Bernacchia or Walton papers discuss how choice stochasticity is related to the reward statistics of the environment. In the Bernacchia paper, the behavioral results center on a characterization of the distribution of RL learning rates (not stochasticity measures), and no attempt is made to examine the relationship of behavior to different task aspects of individual sessions. In the Walton paper, the authors do measure aspects of choice sensitivity, but the results are primarily concerned with differences between controls and lesions and not the relationship between these changes and local reward statistics; these measures are further complicated by the bandit environments, where there is significant variability in values both when option ranks do not change and when they do change (during reversals). Third, as discussed briefly in the paragraph above, neither of the papers uses RL models to examine the relationship between stochasticity and the reward environment. In the Bernacchia paper, RL models are used to fit a behavioral learning rate as a means to compare behavioral and neural time constants of memory in a learning environment. In the Walton paper, RL models are used simply to determine – in addition to objective measures of option values – subjective measures of value. Neither model is intended nor used to compare measures of stochasticity in response to varying reward statistics. In summary, the Bernacchia and Walton papers - while relevant for our findings in terms of the demonstration of long timescales of integration (Bernacchia) and the relationship between OFC and choice stochasticity (Walton) - are difficult starting points for comparison models: they both use reinforcement learning environments, they do not have explicit models for the relationship between choice stochasticity and reward statistics, and the RL models they do use do not present a quantification of the relationship between the reward environment and choice sensitivity.

Finally, it is important to note that we are not claiming our model to be the best model to explain the observed behavior, though we believe that the demonstration of model-behavior correlation explaining adaptation across sessions to be both novel and striking. We view our manuscript primarily as a demonstration that a model that does not explicitly account for statistical information nevertheless reproduces aspects of statistics-dependent behavior (both mean adaptation effect and across-session variability). We believe that this initial manuscript opens the door to future research into whether other models and what types of additional models could reproduce such behavior. While we believe that papers that discuss choice behavior changes in learning environments are relevant, at this point we do not feel – given the differences in task, model construction, and model use outlined above - that they offer suitable models for comparison. Specifically, our animals are in a fully determinate, non-probabilistic choice environment with full information and in which they have

been overtrained; there is no reason to assume that reinforcement learning guides behavior in the task. Therefore, at this time, we have chosen not to include a specific comparison to an RL-type model (and are unsure of what the proper specification would be in this task environment). However, if the Reviewer and editorial staff feels strongly about this point, we would be willing to design and implement a modified RL model for comparison in a further revision.

- Regarding the stats at the top of pg 12 – what data went into this analysis? are these from the slope distributions shown in Fig 3? Or was choice behavior pooled across blocks? Showing the average choice curves in a figure panel would be informative (also for the model predictions in fig 4).

A13R2. We apologize for any lack of clarity in our previous presentation. The Reviewer is correct that the testing refers to the data shown in Fig. 3; essentially we tested whether the mean difference in slopes (between block conditions) was different than zero. Specifically, the statistics are derived from testing the average change in choice curve between the wide and narrow condition over testing days; each daily session provided two data points: choice curve slopes (derived from sigmoid fits) in the narrow and wide blocks. Testing was done with both a parametric paired t-test as well as using nonparametric permutation testing. We opted not to present average choice curves collapsed across conditions because the variability is one of our main points of interest.

To clarify our analysis, we have revised the relevant text in the Results section to read:

“For each session, we fit separate sigmoid functions to test trial choice data from the narrow and wide blocks (see *Methods*). In both monkeys, we observed a significant (Monkey H: $t(38) = 2.63$, $p = 0.012$; Monkey B: $t(29) = 2.19$, $p = 0.036$) change in average choice slope between the two adaptation conditions; this difference was also significant when tested in a non-parametric manner ($p < 0.05$ with permutation testing).”

- Fig 3A : Can the authors bring the individual dots in these panels in front of the fit lines, or otherwise make them more visible? And also clearly distinguish the dots belonging to each condition? This was particularly confusing in the first panel from monkey H, where only the points in the lower part of the plot jump out at first glance. Related to this, it appears that the curve for the ‘narrow’ block in this panel is fit to only points < 0 on the x- axis, so that a sigmoid function isn’t a good fit at all. This raises the possibility that this is not an optimal description of behavior across all sessions, which might indicate that the monkey adopts different strategies in different

blocks. For instance, this one looks like he primarily follows the rule “choose grape” rather than evaluating each offer. Trying to fit behavior that follows a heuristic like this with a model that assumes value-based choices could be misleading – does the model actually produce behavior like this? These data are only from one block, so it’s hard to know how prevalent this is in the results overall. The paper would be more sound if the authors quantify and report sigmoid fits, and consider only modeling those sessions in which sigmoid functions fit the data better than simpler alternatives. It would also be important to understand whether the fits vary systematically across conditions.

A14R2. This is a very valid point that the Reviewer raises, and we appreciate the input on the legibility of Figure 3 and according to the Reviewer’s suggestions have revised the plots to clear up any confusion. Please see the revised Figure 3, where we have: (1) increased the size of the data points, (2) brought them to the foreground, and (3) now distinguish the dots by color to indicate block condition (narrow vs. wide).

We regret that our previous plot did not accurately convey the data, and seek to clear up any confusion here. Because of previous legibility issues, the Reviewer noted that it seemed like data points only occurred at points with $x < 0$ in Figure 3, Monkey H, left plot. However, this is not actually the case as can be seen in the updated figure; the dots in the previous version were hiding behind the curves. In fact, because the x-axis locations were determined by experimentally-set conditions, data are always distributed symmetrically around 0 in the x axis.

However, the Reviewer raises a pertinent point regarding assessing the validity of the choice function fits. We additionally agree with the Reviewer that there could be cases in which, for example, a linear fit might yield a better fit. In the initial submission, we did not perform this comparison; however, what we did do was assess if the Jacobian of the nonlinear solution was in any way ill-defined for any fits. Since the model Jacobian having full column rank equates to $RMSE = \sqrt{(R'R)/(N-p)}$, this is a conservative way to assess data fit quality. [mean RMSE Monkey B = 0.0688, Monkey H = 0.0574]. To address the Reviewer’s concerns in greater detail however, we tested if the average fit varied systematically across our two blocks, something we had not done for the initial submission. We found no significant differences between the RMSE of both the narrow and wide conditions (Monkey B: $t(29) = 1.4058$, $p = 0.1704$, Monkey H: $t(38) = 0.6520$, $p = 0.5183$). This demonstrates that no differences in model fit quality can account for the adaptation-related differences we observed behaviorally.

- Fig 5 shows normalized slopes (narrow – wide), but what about the un-normalized slopes? In other words, are both conditions fit equally well by the model?

A15R2. We thank the reviewer for pointing this out. Since both model and data receive the same normalization, there is only a monotonic scaling that cannot have any effect on the correlation. This procedure is simply done for visualization purposes.

Minor

A16R2. We thank the reviewer for pointing out these minor mistakes. All have been addressed and corrections have been included in the revised manuscript. Please see specific points below.

- pg4, last paragraph – there is an unformatted reference

This reference has now been corrected.

- Fig2 C does not have corresponding text in the figure legend. I assume this is illustrative/hypothetical data, not real data

The Reviewer is correct: the figure is a schematic to depict the hypothesized effect of different reward statistics on value coding neural activity and choice slope data. We apologize for the incomplete figure captioning, and have now revised the figure legend to include the following:

“(C) Illustrative data representing the hypothesized influence of a narrow and wide adaptor distribution on neural coding and choice probability in the test distribution.”

- “test trials” are called “measurement trials” in the methods section. The authors should choose one terminology and use it consistently throughout

We apologize for the lack of clarity, and now refer to these trials as “test trials” throughout the manuscript.

- The methods say “We additionally ran a control regression analysis predicting condition choice stochasticity as a function of number of trials correct, condition completion time, and average response latency.” Where are these results in the manuscript? I think they might be missing

We thank the Reviewer for raising this oversight. This control regression was performed to investigate if other factors that might vary across sessions - such as those related to motivation or attention – could explain the observed across-session variability in adaptation. These analyses revealed no significant relationship between these factors and the extent of across-session adaptation in either animal.

These results are now explicitly stated in the revised Results section:

“This variability in choice stochasticity difference was unrelated to either varying indifference points or other potentially confounding behavioral measures. There was no significant relationship between narrow-wide slope differences and indifference point differences across sessions (Monkey H, $r(38) = 0.09$, $p = 0.56$; Monkey B, $r(29) = 0.20$, $p = 0.30$). Furthermore, a simple regression analysis showed no relationship between narrow-wide choice slope difference and various session variables (percent correct trials, experiment time, average saccade response time, mean received reward, standard deviation of received reward) across sessions (Monkey H, $F(38) = 1.68$, $p = 0.16$; Monkey B, $F(29) = 1.16$, $p = 0.35$).

- Padoa-Schioppa has looked in great detail at choice variability in very similar tasks. A few relevant papers that were not mentioned in this manuscript are Padoa-Schioppa, 2013 (Neuron) and Rustichini et al., 2017 (Nat Comm)

The Reviewer is certainly correct that Padoa-Schioppa’s work is foundational in terms of examining the relationship between reward statistics and neurophysiological and behavioral adaptation during economic choice. The Rustichini paper had not been released at the time of initial submission. We now include references to both the mentioned papers in the revised Manuscript:

“While recent evidence suggests that choice behavior can vary with the range of rewards³⁶ as well as the tendency to repeat choices³⁷ (hysteresis), little is known about the neural mechanisms responsible for adaptive changes in value coding and their potential role in choice behavior.”

“Our results extend these neural findings to the behavioral domain, showing that – consistent with recent theoretical findings³⁶ - value-guided choice behavior also adapts to the recent reward environment.”

Reviewer #3:

The manuscript addresses temporal adaptation phenomena in value based decision making. The authors build on past work, including their own, showing that the neural representation of value adapts to recent reward history. They focus here on two aspects that are less well known: the circuit mechanisms of this reward adaptation, and the link between adaptive changes in the neural coding of value and changes in behavior. The authors do a great job connecting a vast literature on sensory coding, efficient coding, and value based decision making. The experimental results presented in the manuscript support the hypothesis that recent reward history affects choice in non-human primates. The authors also propose a dynamical circuit model that describes well the experimental results, suggesting that temporal statistics of rewards can be captured by dynamic circuit architectures that implement divisive normalization with two separate timescales, and offering a link between the empirical results and the literature on efficient coding. Overall the manuscript reads well, is original, and could influence thinking in the field, but I found some of the claims a bit confusing or based on inaccurate referencing to the literature.

One weakness is that the set up in the abstract and introduction emphasizes the lack of knowledge on the circuit mechanisms underlying of adaptation in the neural representation of value, but the manuscript is not really about circuits: even if the model is circuit-based, it is not meant to suggest a specific circuit implementation in the brain, it is offered simply as a description of the behavioral data. I think this is fine, but perhaps the focus on the abstract could be modified.

A1R3. We thank the Reviewer for this comment and agree that our approach is more of an algorithm/computation approach than a very specific circuit approach even though we model “neurons”. In fact, we believe the Reviewer’s emphasis on the model as a descriptive process of

information processing rather than a specific circuit arrangement aligns closely to our thinking on this matter. The summary of the Introduction now reads:

“These findings indicate that normalization, in our algorithmic and computational approach, can account for both spatial as well as temporal context effects in choice and they suggest that the underlying biophysical mechanism, even though unknown at a circuit level, may serve a common purpose: efficient coding of value representations.”

Another weakness is the proposal that adaptation to reward statistics controls choice sensitivity. The authors suggest this is in agreement with efficient coding ideas, but the literature on sensory adaptation and efficient coding is not uncontroversial: the effects of visual adaptation on discriminability are subtle, stimulus dependent, and varied (compare e.g. Gutnisky and Dragoi 2008; Zavitz et al. J Neurosci 2016; Adibi et al PLoS Comp Biol 2014; for reviews, Kohn and Solomon Curr Biol 2014; Snow et al F1000 2017), Also, contextual modulation (both in time and space) has been shown to induce perceptual biases much more often and more robustly than improvement in discriminability or sensitivity. This is also reflected in models linking contextual modulation, efficient coding, and divisive normalization (e.g. Schwartz et al JoV 2009). Related, on page 4 “Such a retuning of neuronal sensitivities is thought to improve the discriminability (at the neuronal level) between options most likely to be encountered by an organism.” References?

A2R3. The Reviewer addresses an important point regarding the theoretical foundation for the relationship between neural adaptation and choice sensitivity, and we thank them for raising this issue. We apologize for our previous lack of clarity and acknowledge this potential tension in the literature, both here and in revised text in our manuscript. The Reviewer is correct that while the effect of sensory adaptation on discriminability is a long-standing theorized function for adaptation, the empirical evidence for discriminability changes is subtle and a point of contention. Adaptation encompasses multiple phenomena, for example occurring at different timescales and affecting both single neuron firing rates and correlated discharge, and empirical findings are likely organism-, system-, and task-specific. We agree that the link between neural adaptation and behavioral effects is varied, subtle, and to some degree conflicting. To acknowledge this issue, we have revised the relevant text in the Introduction to underscore this contention:

“A common theoretical assumption is that this sensitivity retuning should improve discriminability at the behavioral level; however, despite ample evidence for adaptation effects in sensory neurons, empirical evidence for adaptation-induced changes in perceptual discriminability is varied, subtle, and conflicting^{13,34,35}. In decision making, most previous experiments demonstrating adaptive value coding have not examined choice behavior^{26,33} and the relationship between adaptation in neural coding and changes in empirical decision

making is unclear. While recent evidence suggests that choice behavior can vary with the range of rewards³⁶ as well as the tendency to repeat choices³⁷ (hysteresis), little is known about the neural mechanisms responsible for adaptive changes in value coding and their potential role in choice behavior outside of an explicit learning context³⁸. Thus, a critical open question is whether and how value-based decision making adapts to time-varying changes in the statistics of the reward environment.

Given the unclear relationship between sensory adaptation and changes in discriminability, we agree that there is no a priori reason that changing reward statistics would lead to changes in choice sensitivity. We view this uncertainty as a strength of our manuscript: prior to our experiment, it was unclear whether explicitly changing reward statistics would lead to a corresponding change in choice sensitivity. Thus, our paper provides empirical evidence for a connection between adaptation and improved discriminability (though as the Reviewer notes for sensory adaptation, this effect is subtle, a point addressed in our manuscript Discussion). However, we agree with the Reviewer that adaptation to reward statistics could have additional behavioral effects apart from changes choice sensitivity (analogous to, for example, biases such as repulsion following orientation adaptation). We have now added new text to the revised Discussion section to address the relationship between our empirical findings and the broader literature on perceptual adaptation:

“Our results extend these neural findings to the behavioral domain, showing that – consistent with recent empirical findings³⁶ - value-guided choice behavior also adapts to the recent reward environment. These adaptive changes in choice performance follow in principle from previously demonstrated value coding changes, though establishing a definitive link between adaptation in value coding activity and in choice behavior will require further study. In sensory processing, adaptation does not always reliably improve discriminability for stimuli similar to the adaptor, and often drives other changes in perceptual performance³⁴ (e.g. biases). While our results show a change in value-guided discriminability, adaptation to reward statistics could also produce other changes in choice behavior such as preference biases, particularly under different adaptation conditions.”

I was also concerned that the model captures only a small fraction of the variability in the data (Fig 5, correlation coefficients of .45 and .31). Based on these numbers, claims like “Dynamic normalization accounts for both the average adaptation effect as well as its variability,” should be perhaps be tuned down a bit. More important, I would find it very helpful if the authors could comment on possible sources of the unexplained variability, and whether there are ways to incorporate it in the model.

A3R4. We thank the reviewer for this comment, and it raises several points that we address below. We agree that it is important to recognize that the model clearly does not capture all of the variability in the data, and have as suggested scaled back the strength of the statement in the Discussion:

“Dynamic normalization accounts for both the average adaptation effect as well as a considerable proportion of its variability, suggesting that value adaptation is driven by local reward changes rather than global reward statistics.”

Regarding the model’s ability to capture across-session variability in behavioral adaptation, we agree that the model only captures a portion of the empirical variation across days. However, we believe that the observed correlations are actually quite surprising and represent a good degree of explanatory power considering the nature of the phenomena. The behavior in question – economic choice across good types and amounts – is by its nature stochastic, implying that variability is inherent in the behavior even within sessions; this necessarily means that noise (or variable behavior) is part of the phenomena being modeled, especially when it is propagated to across-session data. Furthermore, as far as we are aware, this is the first attempt to model across session variability with a model that only considers experimental task information, without an explicit term to capture across-session differences. Thus, any other source of behavioral variability – including the factors discussed below, such as motivation and attention – is unavailable to the model. Given this fact, that the model captures a reasonable deal of behavioral variability underscores that the sequence of rewards is an important driver of monkey behavioral adaptation.

The Reviewer is correct that identifying other sources of variability is ultimately essential to fully understanding behavior. Possible sources of across-session variability include nonstationary changes in: preferences, motivation, attention, and satiation, as well as neural/behavioral noise. These factors represent other variables - besides the sequence of rewards and their adaptive effect on value coding as incorporated in the model – that could affect behavior between blocks in a given session or between different sessions. Such factors would be difficult to incorporate into our circuit-based model directly, but one way to examine whether they contribute to the unexplained variability is to determine if there is a relationship between these factors and across-session variability in monkey behavior.

Accordingly, we ran additional regression analyses to see if there was explanatory power in variables such as saccade time, error rate, time in experiment, response times, mean received reward, standard deviation of received reward. None of these factors were significantly related across sessions to the extent of monkey adaptation (empirical difference in choice slopes between narrow and wide blocks), suggesting that they do not contribute to the unexplained variability in Fig. 5A. These control analyses are now included in the Results text as:

“Furthermore, a simple regression analysis showed no relationship between narrow-wide choice slope difference and various session variables (percent correct trials, experiment time, average saccade response time, mean received reward, standard deviation of received reward) across sessions (Monkey H, $F(38) = 1.68$, $p = 0.16$; Monkey B, $F(29) = 1.16$, $p = 0.35$).”

Given the negative results in this control regression, there are several potential explanations for what might contribute to the unexplained variability in the extent of monkey behavioral adaptation. First, we have access to only some select behavioral measurements (e.g. reaction times) and task variables (e.g. mean reward in a session) that serve as proxies for potentially contributing factors such as attention and motivation; it is entirely possible that these are not accurate proxies for these processes or that the processes vary significantly on a finer timescale within sessions. Second, there are potential factors for which we do not currently have accurate proxies, such as satiation and neural noise; such factors remain as potential sources of the unexplained variability. Finally, unexplained variability might arise due to shortcomings of our model: perhaps most or all of the across-session variability could be accounted for by the sequence of rewards, but our model does not use that information in the precise manner as that employed by the monkey brain. We believe that exploring all of these possible explanations are important targets of future work; neurophysiological recording in relevant brain areas (i.e. OFC) will be particularly important in testing and adding additional constraints to the model, a project currently underway in our lab.

I found the lack of alternative models problematic. For instance, are two separate populations/time constants something fundamental, or can this be achieved e.g. with a single recurrent network with heterogeneous time constants? If, as suggested in the Discussion, the author think there is a separation of time constants in different brain areas (LIP and OFC), how much flexibility would be needed for those areas to also capture other time scales different from those presented in these experiments? A smaller, related point is that the choice in the model of a single timescale for both R and G neurons should be motivated.

A3R5. The Reviewer raises an important point, and we agree that a full comparison with alternative models is an important step in evaluating our dynamic normalization model. However, our goal in this paper was to establish – as an initial step - the validity of the multiple-timescale approach: can a simple two-stage circuit with two timescales reproduce adaptive behavior at all? To do so, we examined as a starting point a single model type, where there are two separate circuits with equivalent excitatory and inhibitory time constants within a circuit and different timescales between circuits.

First, regarding the single timescale for R and G neurons, we apologize for the lack of clarity and agree that this is an important assumption to justify; a similar comment was raised by Reviewer 2. The immediate motivation for assuming a single timescale is that we used equivalent excitatory and inhibitory time constants in a previous, single-stage version of the model (Louie et al, 2014). In that paper, a single fast-timescale circuit was shown to accurately describe multiple aspects of LIP value coding activity, and in our extension to a two-stage circuit here we kept the same convention. However, more generally, our rate model is similar in construction to a number of standard models taking mean-field approaches (Wilson-Cowan model and variants; see Wilson & Cowan, 1972, Humanski & Wilson 1992; also see Cowan et al 2016 for review). These models generally assume that excitatory and inhibitory time constants are either equivalent or closely related. To clarify this motivation, we have revised text in the Methods section as follows:

“Excitatory and inhibitory τ values within a given circuit (slow, fast) were set to be equal. Previous work⁴⁵ has demonstrated that a network with equivalent excitatory and inhibitory time constants accurately characterizes value coding activity in decision circuits; more broadly, standard mean-field approaches generally assume equivalent or approximately equivalent timescales within a network⁶¹⁻⁶³.”

We agree with the Reviewer that the separation of the slow and fast timescale circuits of the model into different anatomical regions is not a necessary component or conclusion of our results, and apologize if this was not clear in the previous manuscript. We find it important to note that the monkey-model correlation occurs at specific tau ratios that are roughly similar between animals, suggesting something important about the two timescale structure that we employed. The motivation for different time constants in potentially distinct brain areas stems from recent literature demonstrating longer integration timescales with increased cortical hierarchy; the specific reference to LIP and OFC as potential locales for fast and slow circuit activity was guided by the idea of a separation of action selection and value computation areas. However, it is entirely possible – as pointed out by Reviewer 1 – that other brain regions such as thalamic or striatal areas could play a role, particularly in longer timescale dynamics. In addition, consistent with the Reviewer 3’s intuition, it is entirely possible that the results captured with the two timescale model could be implemented using an entirely different circuit/computation structure such as a reservoir of multiple timescales within a single circuit. We now clarify these issues of interpretation in revised text in the Discussion:

“In our modeling efforts, we purposefully refrained from linking our model stages with particular brain areas; however our estimated slow temporal integration timescales match well with previous electrophysiological studies of adaptation in orbitofrontal cortex and our rapidly adapting stage relates well to data from parietal area LIP^{19,40}. However, it is possible that fast and slow timescale functions are served by a number of brain areas; alternatively, both timescales may be part of a single network capable of operating at a range of multiple timescales. Recent evidence of large-scale dynamical models based on connectivity data from

tract-tracing experiments suggests a hierarchy of integrative timescales with sensory systems exhibiting brief transient responses and persistent long term activity in associative cortex^{44,45}. These findings suggest an unknown circuit mechanism that establishes long temporal receptive windows within prefrontal and temporal areas. One potential explanation for these differences can be regional differences in electrochemical composition of synapses⁴⁶. It is unclear if these synaptic changes are driven primarily within cortical regions or if potential thalamo-cortical projections regulate temporal integration⁴⁷, a potential mechanism underlying many theories of learning signals⁴⁸.”

The issue of whether a single network with heterogeneous timescales could implement this behavior is an important question, but one that we feel is better suited to future work.

In model fits to data, how is the optimal ratio of timescales decided? Based on the peak of 5B? What value is actually used for figure 4? This should be stated more precisely.

A3R6. We apologize for any lack of clarity in the previous manuscript about the timescale ratios. We reiterate that the behavior of the model is governed by the ratio of timescales. The time constant of the fast network is based on findings from previous work (Louie et al. 2014); assuming this value for τ^f (100 ms) allows us to infer a value for τ^s based on model performance at different τ^s/τ^f ratios.(which we can compare to timescales of neurophysiological adaptation in previous findings (Kobayashi et al. 2010, Padoa-Schioppa, 2011)). As inferred by the Reviewer, the optimal ratio was simply identified as the one providing the highest model-monkey correlation across sessions, e.g. the peak in 5B. The data in Figure 4 and Figure 5A were generated using the individual best ratios for each monkey, but the relationship between ratios and model performance (Fig. 5B) was similar in both animals.

The value used reads in the text as: "Figure 4A shows two example model-predicted choice curves in each monkey, for the same sessions displayed in Figure 2 (Monkey B: $\tau^s = 75$ s; Monkey H: $\tau^s = 60$ s)". For the sake of clarity, we now also explicitly include the following statement in the legend for Fig. 4:

“Example model-predicted choice curves in each monkey were based on optimal simulation time scales (Monkey B: $\tau^s = 75$ s; Monkey H: $\tau^s = 60$ s), as shown in Fig. 5B.”

Typos

Fig 1 caption “Panels (B-E) demonstrates” remove *s*

Page 25, first line “(ode45)”?

Page 25, third-to-last line “han,d”

A3R7. We thank the Reviewer for pointing out the typos overlooked in the previous submission. We have addressed them in the revised manuscript.

Reviewers' comments:

Reviewer #1 (Remarks to the Author):

The authors have carefully addressed all of my concerns. I have no further comments.

Reviewer #2 (Remarks to the Author):

I have read the revised manuscript. Overall, I think the additional information provided has improved the study. However, there is one outstanding issue that I feel hasn't been adequately addressed, which is there is no comparison of the proposed model with other potential explanations for the observed choice variability. The author's response to this request gave some perspective on why they chose not to pursue this route, but I think it includes a key misperception. What was observed in the monkeys' behavior is that variability in choices depends on the temporal sequence of recently encountered rewards. Standard ideas in the field would attribute this to learning, most commonly modeled as a simple prediction error-based update function, whereas the authors are attributing it to adaptation implemented through a normalization function. This appears to be based in part on the assumption that there should be no learning in this task, since reward statistics were constant within blocks and the only changes over time were stochastic. However, I don't find it a compelling argument that one can "turn off" reward learning entirely. There are strong ethological reasons for an animal to track and learn about rewards over time, even if fluctuations have been stochastic in the past. Therefore, it's entirely plausible that the monkeys use learning mechanisms to track local reward probabilities and this influences their choices. (One could argue semantics here, that adaptation is a form of learning, but to be clear I'm referring to the standard prediction error-based learning). Therefore, to make a compelling claim that choice variability can be explained with normalization, I think it's critical to show that this is not the same variability that is explained by PE-based learning. The most straight forward way to do this would be to use a temporal-difference learning model to estimate the value of the juice used on adapter trials over time, based on the rewards received and see whether this can predict choice variability in this task.

In addition, one of my previous questions wasn't answered in full: please report the number (or %) of test trials per block – the authors reported only total trials

Finally, I think the information regarding sigmoid fits not varying by condition given in the RTR is relevant and should be included in the manuscript.

Reviewer #3 (Remarks to the Author):

The authors satisfactorily addressed all my comments and suggestions. I find the manuscript overall improved.

I also found the introductory "soliloquy" informative, and would suggest adding some of that information in the Discussion. Something to the extent of 1) model comparison is not the point of this manuscript; 2) nonetheless, model comparison needs to be done, but it is technically challenging for the following reasons, and we leave it for future work.

Reviewer #1:

The authors have carefully addressed all of my concerns. I have no further comments.

A1R1. We thank the reviewer for providing helpful and insightful comments and feel that the manuscript has been significantly strengthened through the reviewer's commitment.

Reviewer #2:

I have read the revised manuscript. Overall, I think the additional information provided has improved the study. However, there is one outstanding issue that I feel hasn't been adequately addressed, which is there is no comparison of the proposed model with other potential explanations for the observed choice variability. The author's response to this request gave some perspective on why they chose not to pursue this route, but I think it includes a key misperception. What was observed in the monkeys' behavior is that variability in choices depends on the temporal sequence of recently encountered rewards. Standard ideas in the field would attribute this to learning, most commonly modeled as a simple prediction error-based update function, whereas the authors are attributing it to adaptation implemented through a normalization function. This appears to be based in part on the assumption that there should be no learning in this task, since reward statistics were constant within blocks and the only changes over time were stochastic. However, I don't find it a compelling argument that one can "turn off" reward learning entirely. There are strong ethological reasons for an animal to track and learn about rewards over time, even if fluctuations have been stochastic in the past. Therefore, it's entirely plausible that the monkeys use learning mechanisms to track local reward probabilities and this influences their choices. (One could argue semantics here, that adaptation is a form of learning, but to be clear I'm referring to the standard prediction error-based learning). Therefore, to make a compelling claim that choice variability can be explained with normalization, I think it's critical to show that this is not the same variability that is explained by PE-based learning. The most straight forward way to do this would be to use a temporal-difference learning model to estimate the value of the juice used on adapter trials over time, based on the rewards received and see whether this can predict choice variability in this task.

A2R1. We thank the Reviewer for their thoughtful explanation and fully agree that variability in choice is often modeled though RL/PE/TD/Q learning models in the literature. We also fully agree that learning cannot be turned off in any case and apologize if our previous response has suggested that we held this view. At a semantic level we do believe that adaptation as implemented with our

normalization approach is a form of learning the statistics of rewards from past outcomes, so we agree with the Reviewer that learning is an ongoing process. But we take the reviewer's point that we need to rule out the possibility that a simple RL model could account for the behavioral pattern we observed. Below we address whether the specific form of learning raised by the reviewer – prediction-error learning implemented as an RL model – can explain the across-session variability in our data (that we argue can be well-captured by the dynamic normalization model).

To address this issue we ran the Reviewer's suggested model on our data. In the standard RL model, the value of each juice type is learned via prediction error as a function of the sequence of experimental rewards. We simply implemented that model (as in Glimcher, 2011) for learning the values of the two reward targets and then implemented a softmax choice function. Two parameters were thus fitted using *fmincon* in Matlab: alpha (the unitary learning rate) as well as beta (the temperature of the softmax function). Parameters were optimized for each individual testing day. The best fitting parameter combination was then used to predict individual choices, and choice curves were fit using the same procedures used in our behavioral and normalization data analyses.

Results of this model performance analysis are shown below in Figure 1. Figure 1A shows scatterplots of the empirically observed adaptation effect (normalized slope difference between narrow and wide conditions) as a function of the adaptation effect predicted by the best-fitting RL model, separately for each animal. *We observed no correlation between RL model and behavioral adaptation variability.* These results demonstrate that the RL model cannot explain the very local temporal effects that the normalization model captures; rather, the RL model captures differences in choice slopes between the two block conditions – events at a much lower temporal scale which are incorporated as static parameters in the faster dynamic normalization model. To ensure that this is not an artificial result stemming from model fitting, we also examined RL model performance while systematically varying the learning rate parameter. Figure 1B demonstrates that this exercise has little influence on improving correlation between adaptation effects in the RL model and empirical behavior, with RL model correlations at all learning rates being much lower than the normalization model correlation values (horizontal lines) in both monkeys; furthermore, none of the RL model correlations are statistically significant ($p > 0.05$ in all cases). These results suggest that the lack of correlation shown in Figure 1A does not arise from the exact RL parameters used. Finally, to quantify the ability of the RL model to explain behavior, Figure 1C shows the goodness of fit (log likelihood) of the RL models with fixed parameters; this shows the ability of the RL models to predict individual monkey choices rather than the across session variability in choice slopes (as in panels A and B). In general, changing the learning rate does not change the ability of the model to predict monkey choices, suggesting that a reinforcement learning process is not the primary driver of monkey behavior in this task (see discussion below).

Taken together, we believe that these results demonstrate that simple RL/TD models cannot account for the stochasticity captured by the dynamic normalization model. We thank the Reviewer for their suggestion to test this approach, we agree that it strengthens our understanding

significantly (and we should have done it in the first place, sorry about that). In any case, it suggests that reinforcement learning is not the primary driver of the observed adaptation and variability effects we focus on in this paper. We believe that this is because, in our task, the reward structure is entirely deterministic: cues signal exactly the amount of reward to be delivered, and rewards are always delivered (no probabilistic outcomes). Thus, an RL model that learns the average value of the juices cannot explain value-driven choices informed by individual trial cues (i.e. choice curves, Fig. 3, manuscript); this is consistent with Fig. 1C shown above, where choice prediction does not depend on RL model learning rate. As the Reviewer suggests, an RL process like this would generate values that depend on past history, with greater variability in values in the wide block condition; but this analysis shows, we believe, that this cannot explain the across session variability in monkey adaptation effects (Fig. 1A and 1B above). To summarize, these analyses show that – in this task – a reinforcement learning process does not explain either monkey choice behavior or across session variability in the extent of adaptation.

We have also included the provided figure into our supplementary materials as Supplementary Figure 2 and added a mention in the main text.

“While reinforcement learning processes in theory can produce sequential trial effects on valuation and generate choice stochasticity, the task used here is entirely deterministic in reward structure and a simple reinforcement learning model does not explain either the observed choice behavior or across-session variability in the extent of adaptation (Supplementary Figure 2).

Figure 1. Reinforcement learning model results. In this standard RL model, the value of each juice type is learned via prediction error as a function of the sequence of experimental rewards, with choice implemented via a softmax choice function. Two parameters were thus fitted using *fmincon* in Matlab: alpha (the unitary learning rate) as well as beta (the temperature of the softmax function). Parameters were optimized for each individual testing day. The best fitting parameter combination was then used to predict individual choices, and choice curves were fit using the same procedures used in our behavioral and normalization data analyses. (A) Scatterplot of the normalized slope differences obtained from the behavioral data versus the normalized slope differences of the reinforcement learning model. For both animals, learning rate (alpha) as well as softmax inverse temperature (beta) were fit independently for each testing day. (B) Reinforcement learning model performance at different fixed learning rates. Dashed lines, correlation between normalization model predictions and empirical observations of adaptation effects (as shown in Fig. 5 in the main text). None of the RL model correlations are statistically significant (all $p > 0.3$). (C)

Boxplots of goodness of fit (log likelihood) of the reinforcement learning for fixed learning rates. Changing the learning rate does not change the ability of the model to predict monkey choices, suggesting that a reinforcement learning process is not the primary driver of monkey behavior in this task.

In addition, one of my previous questions wasn't answered in full: please report the number (or %) of test trials per block – the authors reported only total trials

A2R2. We apologize for the oversight of not having addressed this question. Trial presentation was randomly drawn and was set to result in 60% adapter and 40% test trials. The manuscript has been updated to address this omission. The Methods section now states:

“Trial composition was assigned to be 60 percent adapter trials and 40 percent test trials, with individual trial identity determined randomly.”

Finally, I think the information regarding sigmoid fits not varying by condition given in the RTR is relevant and should be included in the manuscript.

A3R2. We fully agree with the Reviewer and have updated the manuscript accordingly. The manuscript in the Results section now includes:

“We also found no significant difference in the RMSE of the choice curve fits across block conditions (Monkey B: $t(29) = 1.4058$, $p = 0.1704$, Monkey H: $t(38) = 0.6520$, $p = 0.5183$). This demonstrates that differences in model fit quality do not account for the adaptation-related difference in choice performance.”

Reviewer #3:

The authors satisfactorily addressed all my comments and suggestions. I find the manuscript overall improved.

I also found the introductory "soliloquy" informative, and would suggest adding some of that information in the Discussion. Something to the extent of 1) model comparison is not the point of this manuscript; 2) nonetheless, model comparison needs to be done, but it is technically challenging for the following reasons, and we leave it for future work.

A1R3. We express our gratitude to the Reviewer for the help provided. We agree that the manuscript has improved.

REVIEWERS' COMMENTS:

Reviewer #2 (Remarks to the Author):

I have reviewed the additional analyses. The new results significantly improve my confidence in the report, and I have no further concerns.

Reviewer #2:

I have reviewed the additional analyses. The new results significantly improve my confidence in the report, and I have no further concerns.

A1R2. We thank the reviewer for the careful criticism and feedback. We are grateful for the opportunity to strengthen our manuscript.